# SETBP1 accumulation induces P53 inhibition and genotoxic stress in neural progenitors underlying neurodegeneration in Schinzel-Giedion syndrome

Federica Banfi[1,2], Alicia Rubio [1,2], Mattia Zaghi[1], Luca Massimino [1], Giulia Fagnocchi[1], Edoardo Bellini [1], Mirko Luoni[1], Cinzia Cancellieri[1,7], Anna Bagliani[3], Chiara Di Resta[4,5], Camilla Maffezzini[1], Angelo Ianielli[1,2], Maurizio Ferrari[4], Rocco Piazza [6], Luca Mologni[6], Vania Broccoli [1,2] & Alessandro Sessa [1✉]

The investigation of genetic forms of juvenile neurodegeneration could shed light on the causative mechanisms of neuronal loss. Schinzel-Giedion syndrome (SGS) is a fatal developmental syndrome caused by mutations in the SETBP1 gene, inducing the accumulation of its protein product. SGS features multi-organ involvement with severe intellectual and physical deficits due, at least in part, to early neurodegeneration. Here we introduce a human SGS model that displays disease-relevant phenotypes. We show that SGS neural progenitors exhibit aberrant proliferation, deregulation of oncogenes and suppressors, unresolved DNA damage, and resistance to apoptosis. Mechanistically, we demonstrate that high SETBP1 levels inhibit P53 function through the stabilization of SET, which in turn hinders P53 acetylation. We find that the inheritance of unresolved DNA damage in SGS neurons triggers the neurodegenerative process that can be alleviated either by PARP-1 inhibition or by NAD + supplementation. These results implicate that neuronal death in SGS originates from developmental alterations mainly in safeguarding cell identity and homeostasis.

[1] Stem Cell and Neurogenesis Unit, Division of Neuroscience, IRCCS San Raffaele Scientific Institute, Milan, Italy. [2] CNR Institute of Neuroscience, Milan, Italy. [3] Medical Oncology Unit, ASST Ovest Milanese, Legnano Hospital, Legnano, Italy. [4] Vita-Salute San Raffaele University, Milan, Italy. [5] Unit of Genomics for human disease diagnosis, Division of Genetics and Cell Biology, IRCCS San Raffaele Scientific Institute, Milan, Italy. [6] Department of Medicine and Surgery, University of Milano-Bicocca, Monza, Italy. [7] Present address: Human Induced Pluripotent Stem Cells service, Istituto Italiano di Oncologia Molecolare (IFOM), Milan, Italy. ✉email: sessa.alessandro@hsr.it

Diffuse neuronal death, or neurodegeneration, is a pathological landmark of multiple human disorders typically associated with adult life and aging, including Alzheimer's and Parkinson's diseases[1]. Neurodegeneration represents a social burden since millions of people worldwide suffer from symptoms (cognitive decline, memory loss, motor impairment, etc.) associated with the loss of one or more neuronal populations[2]. The pathomolecular mechanisms that result in neuronal cell death and their cell-type specificity remain not fully understood, therefore limiting the current therapeutic options. Genetic cases of neurodegenerative disorders, which often lead to early-onset pathological outcomes, represent interesting entry points to deepen the knowledge of neurodegenerative mechanisms[3]. Some rare pediatric disorders, as the case of neuronal ceroid lipofuscinoses (e.g., Batten disease), present very early degeneration[4] and may further illuminate processes that could (i) have roots in the very initial post-natal life or even embryonically, and (ii) share some mechanisms with the common late forms of neurodegeneration.

Schinzel–Giedion midface retraction syndrome (SGS) (OMIM: 269150) is an ultrarare, devastating condition characterized by multiple congenital malformations, including cardiac, renal, and skeletal abnormalities, neurological impairments, and high risk of cancer[5–8]. Longitudinal studies have suggested early and rapid neurodegeneration to occur particularly in cortical regions in the affected kids[9,10]. Heterozygous de novo point mutations in the SETBP1 gene, leading to the accumulation of its encoded protein, are the sole causes of SGS[11]. All changes leading to classical SGS occurred in a stretch of only 11 nucleotides affecting four consecutive amino acids (D868, S869, G870, and I871) in a degron motif[12,13]. Intriguingly, the somatic counterparts of SGS mutations were discovered in patients affected by atypical Chronic Myeloid Leukemia (aCML) and related diseases[12,14]. In this context, it has been suggested that high levels of SETBP1 protect its interactor, the oncoprotein SET from protease cleavage leading to the formation of a SETBP1-SET-PP2A complex that results in inhibition of PP2A phosphatase activity, thus promoting the proliferation of leukemic cells[13,15,16]. Other than the SETBP1-SET-PP2A axis, diverse SETBP1-mediated mechanisms have been identified as potential oncogenic. In particular, acting as a transcription factor (TF), SETBP1 is able to induce the expression of HOXA9 and HOXA10[14,17]. These are members of the homeobox family of TFs that, in the hematopoietic lineage, are preferentially expressed in hematopoietic stem cells (HSCs), and that are subsequently downregulated as the cell differentiates and matures further in the hematopoietic lineage[18]. Overexpression of these factors results in the enhanced self-renewal potential predisposing myeloid neoplasms. SETBP1 could work as a repressor as part of the NuRD complex causing downregulation of the tumor suppressor RUNX1[19], another TF involved in regulating the differentiation process from HSCs to mature blood cells[20]. Mutant SETBP1 can also activate the well-known oncogene MYB[21], which acts either as a direct activator of oncogenes such as MYC and BCL2 among others or as a repressor of differentiation regulators, such as RUNX1 and CEBPB. Finally, Piazza et al. described SETBP1 as part of the trithorax-like complex contributing to gene expression activation[22]. This study described the SETBP1 binding to DNA, especially into AT-rich regions, with a preference to genome regions nearby the transcription start sites. However, the mechanisms underlying SGS pathology, including the molecular basis of neurodegeneration, remain elusive because of the current lack of appropriate disease models. Human-induced pluripotent stem cells (iPSCs) represent a precious tool to study human genetic conditions[23], in particular in the case of neurological diseases for which primary samples are often not accessible[24]. iPSCs can be derived from patients, thus preserving the genetic architecture that drives the disease, and can be used to generate different derivatives to unravel pathological dysfunctions at multiple levels, i.e., progenitors, differentiated progeny, and organoids[25].

In this work, we generate a human cellular model of SGS with patient-derived and isogenic control iPSCs that enable us to evaluate the impact of SETBP1 accumulation during neuronal commitment and maturation. We uncover cancer-like features in SGS neural progenitor cells (NPCs) including increased cell proliferation and accumulation of DNA damage associated with P53 hypoactivity, without evident alterations in the PP2A function. Our results indicate that the inheritance of DNA damage during neuronal development predisposes postmitotic neurons to early cell death through a PARP-1-dependent mechanism.

## Results

**SGS iPSCs do not display signs of SETBP1 accumulation.** To study the pathological effects of SGS-causing SETBP1 mutations in a human in vitro model, we reprogrammed fibroblasts obtained from two SGS patients and two age-matched controls (WT1 and WT2) into iPSCs through the Sendai virus non-integrant method (Fig. 1a). Among the SGS patients, one carries the isoleucine (I) to threonine (T) substitution in position 871 (I871T), while the other one has an aspartic acid (D) to asparagine (N) substitution in position 868 (D868N)[11] (Fig. 1a). To minimize uncontrolled genetic or epigenetic variability due to interindividual differences[26], we corrected the SETBP1 mutations obtaining isogenic control iPSCs (I871I and D868D) by means of CRISPR/Cas9 technology (Supplementary Fig. 1a and Fig. 1a). No alterations in predicted off-target genes were retrieved in the edited cell lines (Supplementary Fig. 1a). All the selected iPSC lines for this study presented a normal karyotype, high levels of pluripotency markers, and multilineage differentiation capability (Supplementary Fig. 1b, c).

Because SGS mutations cause SETBP1 accumulation[13], we assessed SETBP1 protein levels by western blotting on total lysates of undifferentiated iPSCs. Surprisingly, we did not find any differences between SGS cells and controls (Fig. 1c). Also, SETBP1 mRNA levels were comparable among genotypes (Supplementary Fig. 1d), indicating that the expected accumulation was not blunted by compensation at the transcriptional level. Accordingly, we retrieved neither accumulation of SET protein (or of its RNA) (Fig. 1d and Supplementary Fig. 1e) nor PP2A activity deficiency as assessed by the ratio of the phosphorylated form (Tyr307) on total PP2A and direct measurements of phosphatase activity (Fig. 1e, f). Mutant iPSCs displayed a normal proliferation rate as assessed by the count of mitoses using phospho-histone H3 (pH3) immunostaining (Fig. 1g and Supplementary Fig. 1f).

These results indicate that SGS IPSCs are indistinguishable from their wild-type counterpart, at least at the level of basic properties (e.g., self-renewal, differentiation, proliferative capability) likely because SETBP1 degron mutations do not exert any change in SETBP1 protein level at this early developmental stage.

**SGS NPCs accumulate SETBP1 and overproliferate.** Since the strong neurological alterations afflicting the SGS patients, we sought to derive NPCs from control and SGS iPSC lines. Adapting a small-molecule-based multistage protocol using small molecules[27], we obtained a homogeneous population of neural progenitors (NESTIN+ and SOX2+) from all genotypes with comparable yield and cortical identity (FOXG1+ and PAX6+) (Fig. 2a, b and Supplementary Fig. 2a).

We investigated the eventual accumulation of SGS-associated proteins at the NPC stage. Both biochemical and immunocytochemical

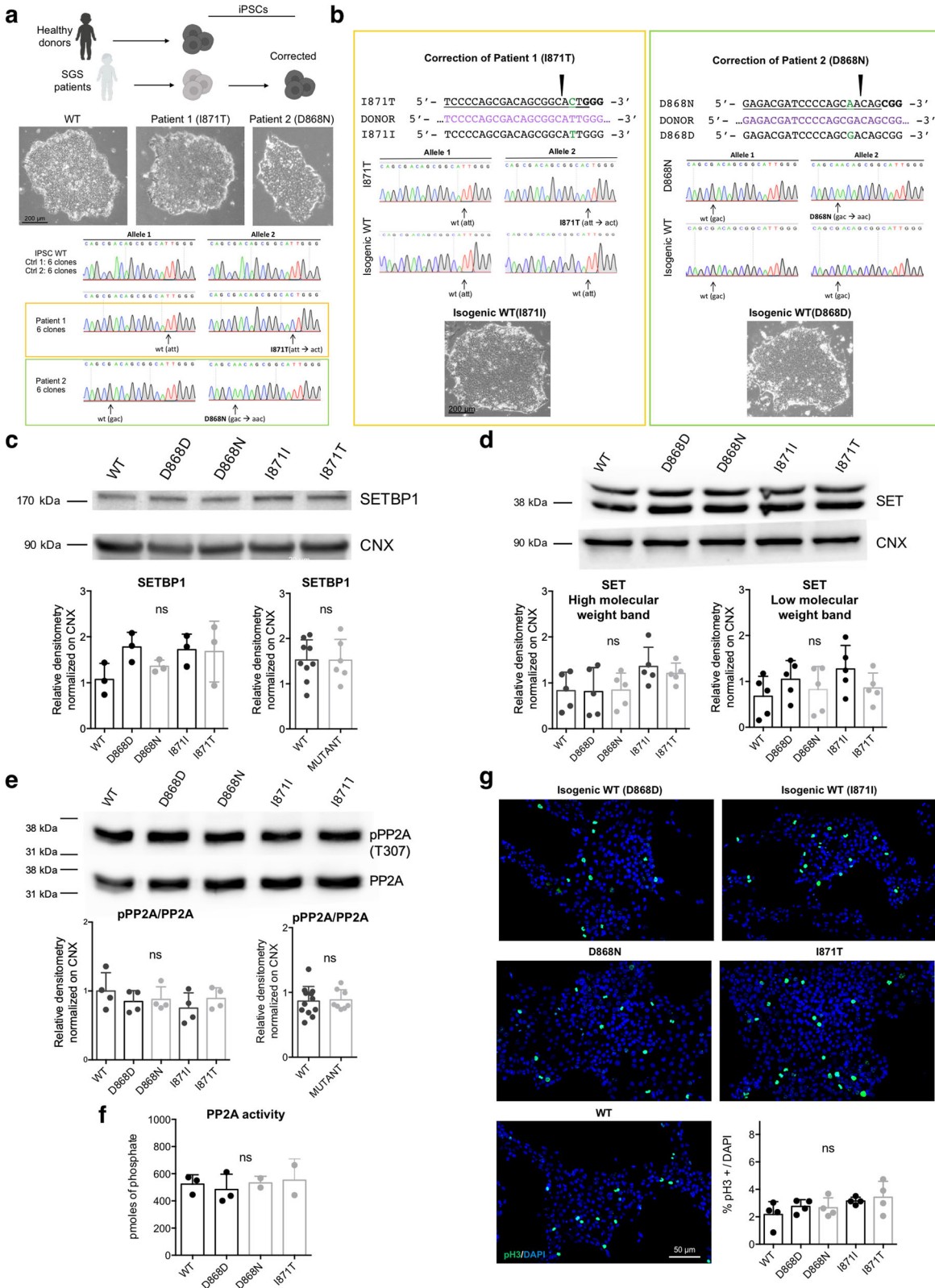

analyses indicated that SGS mutations on the *SETBP1* gene led to the accumulation of the protein, which was not mirrored by a corresponding difference at the mRNA level (Fig. 2c and Supplementary Fig. 2c). Of note, we also found that also SET accumulates in SGS NPCs (Fig. 2d and Supplementary Fig. 2b) probably as a consequence of SETBP1 stabilization. This hypothesis was further strengthened by the observation that the levels of SET mRNA were comparable among

genotypes (Supplementary Fig. 2d). However, PP2A phosphatase activity resulted unchanged in SGS NPCs (Fig. 2e, f and Supplementary Fig. 2e), indicating that the increased levels of SETBP1 and SET may have different molecular outputs in neural cells than in the myeloid lineage[15,16].

At the phenotypic level, SGS neural progenitors showed a robust increase in the proliferation rate compared to both

**Fig. 1 SGS iPSCs do not display of SETBP1 accumulation. a** Fibroblast reprogramming from age-matched healthy donors (2) and SGS patients (2) and correction of patient-derived iPSCs (upper panel). Representative bright-field images (taken at the same magnification) of iPSC colonies derived from a healthy donor and SGS patients, (middle panel). Sanger sequencing confirmed the presence of the indicated mutations (lower panel, $n = 6$). **b** Back-mutation strategy in patients-derived iPSCs. For each patient, the mutation (green aa) was targeted with CRIPSR/Cas9, and the correction was confirmed by Sanger sequencing (correction efficiency: 1/48 clones for pt.1, 1/24clones for pt.2). Representative bright-field images (taken at the same magnification) of colonies from both isogenic corrected iPSC lines. **c** SETBP1 immunoblotting in isogenic control (D868D and I871I), mutant (D868N and I871T) and WT iPSCs and relative quantification. D868D vs. D868N $P = 0.6973$, I871I vs. I871T $P > 0.9999$; WT vs. MUT $P = 0.9798$. $n = 3$. **d** ET immunoblotting in isogenic control and mutant iPSCs and relative quantification. See Supplementary Dataset 3 for details of statistical analysis and Supplementary Fig. 1f for WT vs. MUT comparisons. $n = 5$. **e** Total and phosphorylated (pPP2A Tyr307) PP2A immunoblotting in isogenic control and mutant iPSCs and relative quantification of pPP2A/PP2A ratio. WT vs. D868D $P = 0.8163$; WT vs. D868N $P = 0.9085$; WT vs. I871I $P = 0.4326$; WT vs. I871T $P = 0.9383$; D868D vs. D868N $P = 0.9994$, I871I vs. I871T $P = 0.8531$; WT vs. MUT $P = 0.8363$. $n = 4$. **f** PP2A phosphatase activity in isogenic control and mutant iPSCs. D868D vs. D868N $P = 0.9487$, D868D vs. I871I $P = 0.8743$, WT vs. I871T $P = 0.9882$ $n = 3$. **g** Phosphorylated H3 (pH3 Ser10) immunostaining in isogenic control and mutant iPSCs and quantification. D868D vs. D868N $P = 0.9998$, I871I vs. I871T $P = 0.9868$, see Supplementary Fig. 1g for WT vs. MUT comparison. $n = 4$. All data expressed as mean ± SEM. ns: nonsignificant differences when $P > 0.05$. Images taken at the same magnification. **c–g** One-way ANOVA followed by Tukey post hoc test for multiple comparisons, except for WT vs. MUT comparisons two-sided unpaired $t$ test in **c** and **e**.

isogenic and unrelated wild-type NPCs, as suggested by growth curve analysis over several passages in culture (Fig. 3a) and by mitoses count through pH3 mitosis marker immunostaining (Fig. 3b and Supplementary Fig. 2f). Overproliferation of mutant NPCs was further confirmed by cell cycle analysis using propidium iodide staining followed by fluorescence-activated cell sorting (FACS). Accordingly, the peak analysis of FACS graphs showed a reduced fraction in the G1 phase, and a corresponding higher percentage in S and M/G2 phases in of SGS NPCs compared to wild-type cells, in accordance with their more active proliferative kinetics (Fig. 3c and Supplementary Fig. 3a). To better characterize the cell cycle progression of both control and SGS cells, we analyzed by flow cytometry the G1/S exit after thymidine double block at different time points. The peak analysis of these FACS graphs showed that mutant cells were faster in exiting from G1 to reach S and G2 phases (Supplementary Figs. 2g and 3b).

Together, these findings show that the *SETBP1* mutations lead to the expected SETBP1 and SET accumulation in SGS NPCs, indicating that our cellular system is adequate to study the neural aspects of SGS. Of note, SGS NPCs overproliferate, consistently with what observed in myeloid cells with *SETBP1* overexpression, but PP2A inactivation does not seem to represent a key pathological event in the neural lineage.

**SGS mutations lead to P53 inhibition and oncogenic signature**. To thoroughly analyze SGS-specific molecular alterations, we performed transcriptomic analysis in diseased and control NPCs. SGS NPCs displayed massive transcriptomic changes with thousands of transcripts both up- and downregulated after SETBP1 accumulation (FDR < 0.01) (Fig. 4a and Supplementary Dataset 1). Many deregulated genes were in common between the two different SGS genotypes analyzed (Supplementary Fig. 4a and Supplementary Dataset 1). Gene enrichment analysis (Gene Ontology, GO) indicated that mutant NPCs have reduced expression levels of genes related to mitotic cell cycle arrest, response to misfolded protein, and regulation of morphogenesis among others, while increased the transcriptional levels of genes involved in several pathways related with cell proliferation in general and in NPCs in particular, including WNT and Notch signaling and MAP kinase cascade (Fig. 4b and Supplementary Dataset 2). Intriguingly, mutant NPCs upregulated genes related to cell lineages with a very different developmental origin, for instance, genes related to the inflammatory response (*IL1B*, *CSF1*, *ADORA1*, *IL1R*, *SPP1*) and angiogenesis (*THY1*, *RAMP1*, *ADM2*) among others (Fig. 4c, Supplementary Fig. 4b and Supplementary Dataset 2). However, an in silico analysis using the t-distributed stochastic neighbor embedding (t-SNE) algorithm indicated that

mutant and wild-type NPCs correlated with each other more closely and strongly, in respect to other iPSC-derived cell types (Fig. 4d). Thus, these results confirmed that, despite showing a dysregulated molecular signature, mutant NPCs still retained strong neural progenitor identity.

Consistent with the previously detected alterations in proliferative capacity (Fig. 3), we found a profound dysregulation of genes associated with the cell cycle in SGS NPCs (Supplementary Fig. 4c). An in-depth analysis of differentially expressed genes revealed that SGS NPCs strongly upregulated several elements of both the MAPK cascade, cyclins, cyclin-dependent kinases, and the family of E2Fs transcription factors (Supplementary Fig. 4c–e).

We detected several deregulated transcripts among cancer-related genes, either activated oncogenes (*MYC*, *AKT2*, *FOS*, *BCL11A*, *EGFR* among others) or downregulated oncosuppressors (*BRCA2*, *CDKN2C*) (Fig. 4e, f and Supplementary Fig. 4f) further strengthening the possibility that mutant progenitors were deeply transformed when compared to the wild-type.

As independent evidence for this, we found that the key tumor suppressor *PTEN* gene, which is associated both with many cancer types including glioblastoma and with a larger brain volume (megalencephaly)[28], was significantly downregulated in SGS NPCs (Fig. 4f and Supplementary Fig. 4f). *PTEN* encodes for the phosphatidylinositol-3,4,5-trisphosphate (PIP$_3$) phosphatase, which was as well downregulated in our SGS NPCs (Fig. 4g), and which negatively regulates intracellular levels of PIP$_3$ in many cell types, including neural progenitors[29]. A decrease in PIP$_3$ results in the inhibition of the AKT/PI3K signaling pathway that is important for the regulation of cell growth, proliferation, and survival[30]. In accordance with these evidence, SGS NPCs exhibited hyperactivation of the AKT pathway as indicated by its increased phosphorylation level (phosphoT308 and phosphoS473) and by the upregulation of its downstream effectors, including phosphorylated S6 (phospho S235/236) and mTOR (phospho S2448), while NF-κB remained unchanged (Supplementary Fig. 5a–c). To better dissect the importance of the AKT pathway, we used the selective inhibitor MK-2206[31] which was able to reduce both AKT and pS6 levels and to normalize the number of mitotic figures in NPC mutant cultures (Fig. 4h–j).

In line with the hyperproliferative features and the loss of transcriptional control of oncogenes and oncosuppressors, we observed that our mutant neural progenitors displayed a tendency to accumulate DNA damage, as testified by the increase of pH2AX (phosphoS139) in these cells. pH2AX is a marker of nuclear foci where DNA double-strand breaks (DSBs) occur and DNA damage repair (DDR) initiates and was quantified both by counting the number of dots per cell (Fig. 5a) as well as by its relative quantity, measured by western blotting (Fig. 5b).

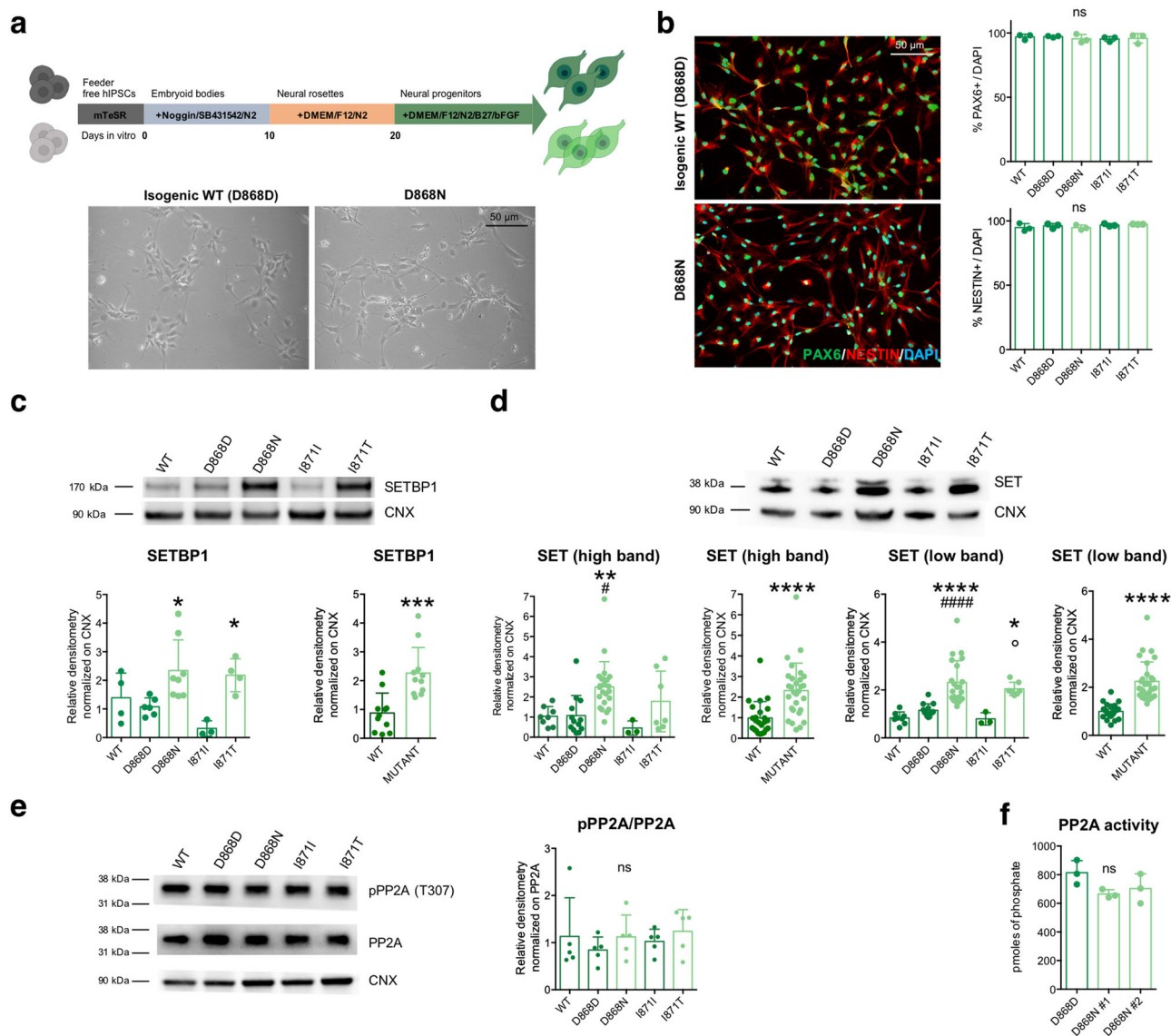

**Fig. 2 SGS NPCs display features of SETBP1 accumulation. a** NPC generation protocol (upper panel) and representative bright-field images of isogenic control (D868D) and mutant (D868N) NPCs. **b** Representative images (taken at the same magnification) of NESTIN (P = 0.3897) and PAX6 (P = 0.8436) immunostaining and relative quantification. n = 3. See also Supplementary Fig. 2a. **c** SETBP1 immunoblotting in WT, isogenic control (D868D and I871I) and mutant (D868N and I871T) NPCs and relative quantification. * indicates statistic test between isogenic cell line pairs. P = 0.0048; WT vs. D868D P = 0.9686; WT vs. D868N P = 0.2983; WT vs. I871I P = 0.3912; WT vs. I871T P = 0.6119; D868D vs. D868N *P = 0.0465; I871I vs. I871T *P = 0.0361; D868N vs. I871T P = 0.9966. WT vs. MUT ***P = 0.0004. n = 3. **d** SET immunoblotting in WT, isogenic control and mutant NPCs and relative quantification * indicates statistic test between isogenic cell line pairs. # refers to the comparison between each mutant cell line with the unrelated WT. ° refers to the comparison between mutant NPCs. n = 3. See Supplementary Dataset 3 for details of statistical analysis. **e** r total and phosphorylated PP2A (pPP2A Tyr307) immunoblotting in WT, isogenic control and mutant NPCs and relative quantification of pPP2A/PP2A ratio. P = 0.7713. D868D vs. D868N P = 0.8981; I871I vs. I871T P = 0.9576; See Supplementary Fig. 2e for WT vs. MUT comparisons. n = 5. **f** Phosphatase activity assay of PP2A in isogenic control and mutant NPCs. P = 0.1343, D868D vs. D868N #1 P = 0.1315, D868D vs. D868N #2 P = 0.2760. #1 and #2 refer to two independent clones of the D868N IPSC line. n = 3. All data expressed as mean ± SEM. ns: nonsignificant differences when P > 0.05; *P < 0.05, **P < 0.01, ***P < 0.001, ****P < 0.0001. **b–f** One-way ANOVA followed by Tukey post hoc test for multiple comparisons, except for WT vs. MUT comparisons two-sided unpaired t test in **c** and **d**.

Accordingly, other markers of early DDR events resulted to be increased in SGS neural progenitors, such as phosphorylated ATM (phosphoS1981) and CHK2 (phosphoT68) (Supplementary Fig. 6a). Alkaline single-cell gel electrophoresis (comet assay) revealed that also DNA damage levels, per se, were augmented in mutant cells (Fig. 5c), suggesting that, despite the higher DDR activation, either the damages were not properly resolved or that, at least, that new breaks were created faster than they were repaired. Interestingly, the accumulation of DNA damages did not result in the activation of the apoptotic response. Apoptosis activation was tested by immunostaining for cleaved caspase 3, one of its key executioners (Fig. 5d) and FACS analysis (Supplementary Fig. 6b, population with DNA content lower than 2 N is highlighted in yellow), finding no evidence of activation of programmed cell death in mutant NPCs, as was indicated at transcriptomic level (Fig. 4b). Thus, mutant cells displayed DNA damage, without neither apoptosis activation nor cell cycle pausing. On the same line, we observed both an

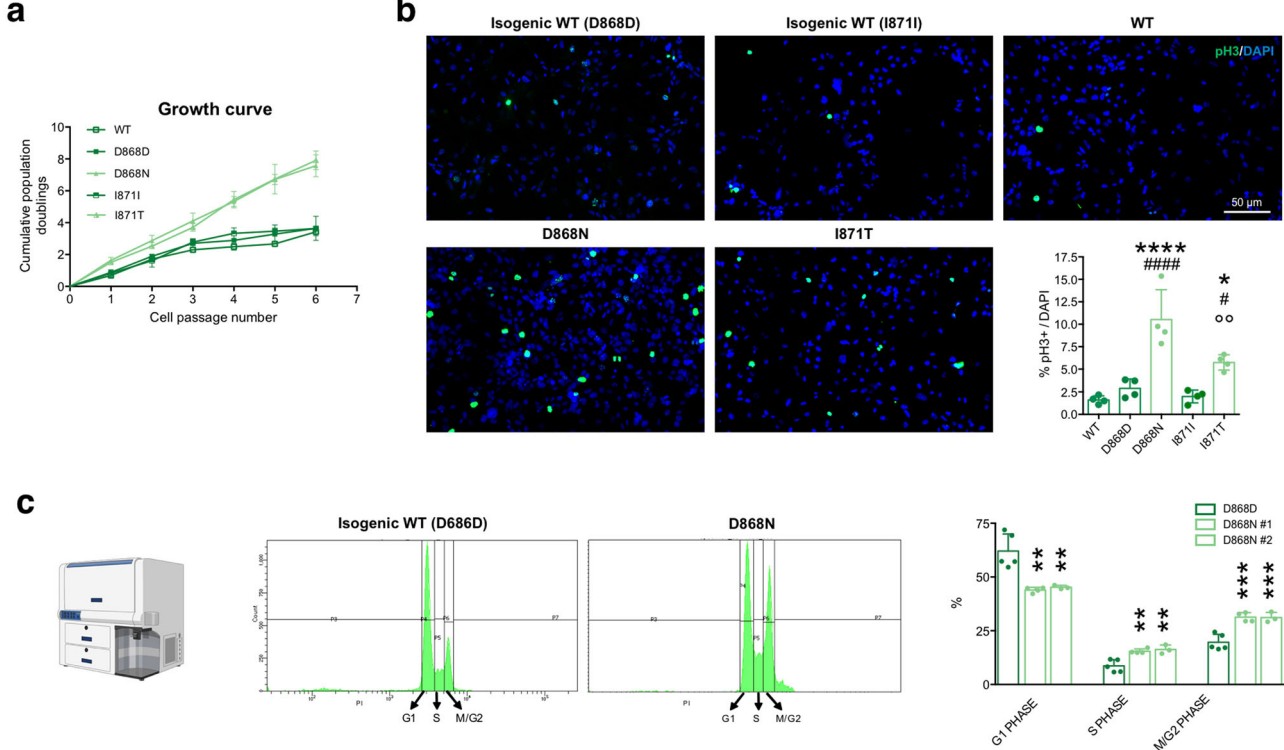

**Fig. 3 SGS NPCs display overproliferation. a** Growth curve analysis of WT, isogenic control, and mutant NPCs See Supplementary Dataset 3 for statistical analysis. $n = 3$. **b** NPC proliferation assay by phosphorylated H3 (pH3 Ser10) immunostaining and quantification in WT, isogenic control, and mutant NPCs. * indicates statistic test between isogenic cell line pairs, # refers to the comparison between each mutant cell line with the unrelated WT one, ° refers to the comparison between mutant NPCs. $P < 0.0001$; WT vs. D868D $P = 0.8012$; WT vs. D868N ####$P < 0.0001$; WT vs. I871I $P = 0.9972$; WT vs. I871T #$P = 0.0202$; D868D vs. D868N ****$P < 0.0001$; I871I vs. I871T *$P = 0.0377$; D868N vs. I871T °°$P = 0.0068$. See Supplementary Fig. 2f for WT vs. MUT comparisons. $n = 4$. Images taken at the same magnification. **c** Cell cycle analysis by PI staining and FACS quantification; graphs showing different peak distribution indicating different cell cycle phases in isogenic control and mutant NPCs. Bar graph shows the percentage of cells present in each phase in D868D and D868N cells. G1: D868D vs. D868N #1 ****$P = <0.0001$, D868D vs. D868N #2 ****$P < 0.0001$; S: D868D vs. D868N #1 *$P = 0.0311$, D868D vs. D868N #2 *$P = 0.0253$; M-G2: D868D vs. D868N #1 ***$P = 0.0002$, D868D vs. D868N #2 ***$P = 0.0007$. $n = 3$ independent experiments. All data expressed as mean ± SEM. ns: nonsignificant differences when $P > 0.05$; *$P < 0.05$, **$P < 0.01$, ***$P < 0.001$, ****$P < 0.0001$. **a**, **c** Two-way ANOVA followed by Tukey post hoc test for multiple comparisons, **b** one-way ANOVA followed by Tukey post hoc test for multiple comparisons.

anomalous size and shape of nuclei and the presence of multinucleated and polyploid cells with DNA content higher than 4 N in mutant cultures (Supplementary Figs. 6c and 3a), and further confirmed this result by PI staining and FACS analysis (Supplementary Fig. 6b, highlighted in red).

Recently[32], suggested that the acidic domain present in SET protein can bind the lysine-rich domain of other proteins when the lysines are not acetylated[32]. Through physical hindrance to the acetylation, this binding would either inhibit or delay the subsequent activation of a group of proteins of which P53 is the main example[32]. We wondered whether SGS mutations that increase SET levels, could affect P53 activity. To address this point, we first evaluated the fraction of P53-acetylated form (acetylLys382), which resulted thickly diminished in SGS NPCs, while the total protein amount appeared unchanged (Supplementary Fig. 6d). Moreover, we treated control and SGS NPCs, with doxorubicin, an anthracycline widely used to treat leukemia, to induce acute DNA damage response, of which P53 activation is part. Remarkably, while the antibiotic was equally able to increase pH2AX levels in both genotypes (Supplementary Fig. 6e), SGS NPCs did not increase their active P53, as shown by its acetylated (K382) and phosphorylated (S15) forms (Fig. 5e). Finally, we employed the small molecule Nutlin-3a during SGS NPC derivation to disrupt P53-MDM2 interaction and impede the P53 degradation, thus, favoring the activation of the pathway

without genotoxic stress[33]. As expected, the Nutlin-3a treatment increased the level of P53 protein as well as the amount of its acetylated form, albeit the ratio between the two remained very low (Fig. 5f). The P53 enhanced levels protected the progenitors from the DNA damage accumulation that was observed in the SGS background (Fig. 5g and Supplementary Fig. 7a) and normalized proliferation defect (Supplementary Fig. 7b, c), while apoptosis was unchanged (Supplementary Fig. 7d). Of note, cell cycle-related genes that were deregulated in SGS genotype (*CCND1*, *CCND3*, *MYC*, *CCNG1*) were only partially rescued by the modulation of P53 levels (Supplementary Fig. 7e), suggesting that a P53-independent control of transcription was still in place, possibly as a direct consequence of SETBP1 or SET accumulation.

Altogether these data unveiled cancer-like deficits in SGS NPCs with loss of cell proliferation control, DNA damage accumulation, hypoactivation of P53 signaling, and lack of apoptosis activation, associated with dysregulation of genes and pathways related to tumorigenesis.

**Mutant organoids recapitulate SGS pathological features.** Motivated by our findings in 2D in vitro neural cultures, we sought to evaluate SGS pathological signs in 3D organoid structures that better recapitulate the brain complexity and its

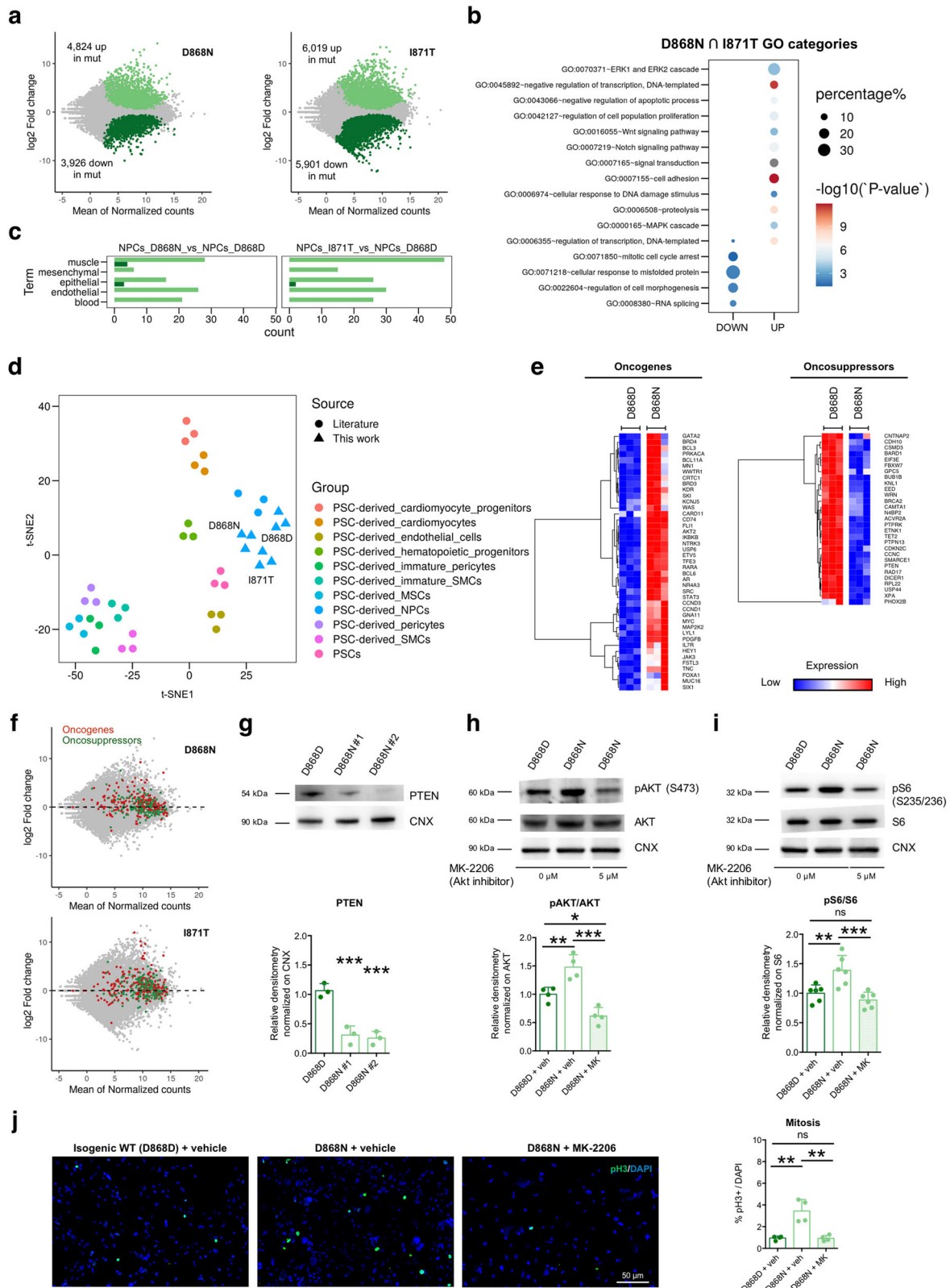

multicellular composition. Thus, starting from control and SGS iPSCs we generated both cerebral organoids[34] (Fig. 6a) and cortical spheroids[35] (Supplementary Fig. 9a).

SGS cerebral organoids presented SETBP1 and SET protein accumulation which did not correlate with PP2A activity inhibition, as observed in 2D NPC cultures (Fig. 6b). At the morphological level, mutant organoids grew in a more elaborated

manner when compared to the wild-type (WT) ones, presenting large convolutions and a more folded and expanded neuroepithelium (Fig. 6c, d and Supplementary Fig. 8a). Immunofluorescence analyses showed neuroepithelial-like structures with typical cell-to-cell junctions (ZO1+) among progenitors of cortical identity (NESTIN+ and PAX6+) in both genotypes with SGS organoids displaying less ZO1+ rosettes but significantly

**Fig. 4 SGS mutations lead to profound gene dysregulation and oncogenic signature. a** MA plot showing log2-fold changes as a function of average gene expression for isogenic pairs of NPCs. Differentially expressed genes highlighted with their respective color codes. **b** Enrichment plot displaying relative enrichment (%; size) and statistical significance ($-\log10$ P value; color scale) of gene ontology categories of interest in SGS lines. **c** Histograms showing the number of GO categories relative to the indicated different lineages, calculated in the list of genes upregulated in controls (dark green) and mutant NPCs (light green) as in RNA-seq dataset. **d** Whole transcriptome dimensionality reduction by t-SNE highlighting the similarities between D868D, I871T, and D868N NPCs in comparison with other pluripotent stem cell derivatives (data from literature), including NPCs. **e** Gene expression heatmaps of upregulated oncogenes and downregulated oncosuppressors in D868N vs. D868D NPCs. **f** COSMIC Oncogenes (red) and oncosuppressor genes (green) among the transcriptomic profiles of the two mutations, shown as highlighted dots in the MA plots. **g** PTEN immunoblotting in isogenic control and mutant NPCs (#1 and #2: independent clones): D868D vs. D868N #1 ***P = 0.0010, D868D vs. D868N #2 ***P = 0.0007, n = 3. **h** Total and phosphorylated AKT (pAKT Ser473) immunoblotting in isogenic control (+vehicle) and mutant NPCs (+vehicle or MK-2206 AKT inhibitor) and quantification as pAKT/AKT ratio. D868D + veh vs. D868N + veh **P = 0.0079, D868D + veh vs. D868N + MK *P = 0.0261, D868N + veh vs. D868N + MK ***P = 0.0001, n = 4. **i** Total and phosphorylated S6 (pS6 Ser235 and pS6 Ser236) immunoblotting in isogenic control (+vehicle) and mutant NPCs (+vehicle or MK-2206) and quantification as pS6/S6 ratio. D868D + veh vs. D868N + veh **P = 0.0063, D868D + veh vs. D868N + MK P = 0.5338, D868N + veh vs. D868N + MK ***P = 0.0007. n = 6. **j** NPC proliferation assay by phosphorylated H3 (pH3 Ser10) immunostaining in isogenic control (+vehicle) and mutant NPCs (+vehicle or MK-2206). D868D + veh vs. D868N + veh ***P = 0.0009, D868D + veh vs. D868N + MK P = 0.9990, D868N + veh vs. D868N + MK ***P = 0.0009. n = 4. Data expressed as mean ± SEM. *P < 0.05, **P < 0.01; ***P < 0.001, ****P < 0.0001. Images taken at the same magnification. **g–j** Data are analyzed by one-way ANOVA followed by Tukey's post hoc test for multiple comparisons.

longer (length of the lumen) and thicker than their WT counterpart (Supplementary Fig. 8b–d). Immunochemistry for KI67, a marker of cell proliferation, and MAP2, labeling differentiated neurons, revealed an increase in neural progenitor proliferation and a relative loss of neuronal production (Fig. 6e). This can well explain the excessive neuroepithelial layer expansion responsible of the folded appearance of mutant organoids, thus confirming our previous findings in 2D cultures. Our data strikingly resemble the folded phenotype observed in organoids derived from cells in which the PI3K/AKT axis was enhanced by loss of PTEN function[31]. Interestingly, AKT signaling was increased also in SGS organoids (Fig. 6f) as already demonstrated in the 2D setting.

Then, we profiled the single-cell transcriptome of 4-week-old organoids from both isogenic control (D868D) and SGS cells (D868N). Cells from both genotypes were well intermixed in the uniform manifold approximation and projection (UMAP)[36], suggesting a high grade of concordance in the organoid identity (Fig. 6g). k-nearest neighbors based clustering of organoids-derived single cells identified ten clusters (Fig. 6g, Supplementary Fig. 8e and Supplementary Dataset 1). From the expression of gene markers, we combined the identified clusters into seven major cell categories: early progenitors (e.g., SOX2+, PAX6+, KRT8+, DLK+) (neuroepithelial cells, NEPs), neural stem cells (e.g., SOX2+, PAX6+, PCNA+, MKI67+) (NSCs), progenitors with less proliferative markers that we named as quiescent NSCs (e.g., NES+, SOX2+, PAX6+, PCNA^low, MKI67−, PTPRZ1+) (qNSCs), progenitors with the expression of both proliferative and differentiation markers (e.g., SOX2+, FABP7+, CCND2+, PTPRZ1+) (difNSCs), immature neurons (e.g., DCX+, NHLH1+) (immNs), neural cells displaying a mix of markers of several stages from different brain regions (e.g., MSX1+, RSPO3+, PCP4+, CXCL14+) and unclassified cells (Fig. 6h and Supplementary Fig. 8f). On the basis of this annotation, organoids of both genotypes consisted mainly of progenitor cells (e.g., expressing SOX2 and PAX6) as expected after 1 month of differentiation[37] (Fig. 6h). Interestingly, SGS organoids showed collectively more proliferating progenitors, which expressed markers of cancer-related proliferative genes (e.g., CRABP2, EGFR) virtually absent, or low in controls (Fig. 6h and Supplementary Fig. 8g). In particular, SGS organoids were enriched of progenitors of misspecified identity (difNSCs), immature neurons, and depleted of cells from other brain areas compared to control (Fig. 6h and Supplementary Fig. 8g). In accordance with our 2D data (Fig. 3c), the distribution of cell cycle stages of SGS organoid cells was suggestive of enhanced proliferation, with a higher number of cells in the S phase and less in the G1 phase compared to control (Fig. 6i).

Next, we investigated the eventual presence of unresolved DNA damage in a 3D context. We found increased levels of pH2AX in mutant compared to wild-type organoids (Fig. 6j, k), again confirming the findings in NPC cultures. To better investigate this phenotype, we also analyzed more mature organoids. After 8 weeks of differentiation, SGS organoids maintained their folded appearance (Supplementary Fig. 8h) and the accumulation of DNA damage in both SOX2+ progenitors and MAP2+ postmitotic neurons (Supplementary Fig. 8i). Accordingly, mutant compared to wild-type organoids presented a higher number of positive cells to the TUNEL assay that labels DNA double-strand breaks in both living cells with DNA damage and dying cells (Supplementary Fig. 8j).

On the other hand, SGS cortical spheroids remained smooth possibly indicating a requirement for non-dorsal cells for folding formation. However, in line with our previous results, they were bigger than the corresponding controls, possibly because of an overproliferation of neural progenitors (Supplementary Fig. 9b, c). As detected in organoids and 2D NPCs, SGS spheroids also presented accumulation of SET, which did not correlate with PP2A activity inhibition (Supplementary Fig. 9d) and increased levels of pH2AX, when compared to the control counterparts (Supplementary Fig. 9e).

These findings provide additional evidence on the general SGS pathological features, including neural progenitor overproliferation and DNA damage accumulation, both equally detectable in multicellular 3D structures and NPC cultures.

**SGS neurons inherit excessive DNA damages causing degeneration.** To reconcile with the brain pathology described in SGS patients, which affects postmitotic neural cells in patients[9,10], we generated cortical neurons carrying either SGS mutations or the corresponding corrected alleles, by forcing the expression of the pro-neural factor NEUROGENIN 2 in NPCs[38] in a chemically defined medium[39] (Fig. 7a). We obtained neurons from both lines, although mutant neurons displayed some decrease in the total yield of MAP2+ cells (Fig. 7b) and a simpler morphology, as assessed by Sholl analysis (Supplementary Fig. 10a). SGS neurons showed accumulation of SETBP1 and SET, without any difference in PP2A activity as previously shown in NPCs and organoids (Supplementary Fig. 10b, c). Albeit neurons are postmitotic cells with less chances of experiencing acute DNA damage, SGS neurons might have inherited unsolved genomic damages derived from their respective cells of origin. In agreement with this hypothesis, SGS neurons showed a particularly enriched pH2AX staining (Fig. 7c and Supplementary Fig. 10d). We reasoned that the excessive

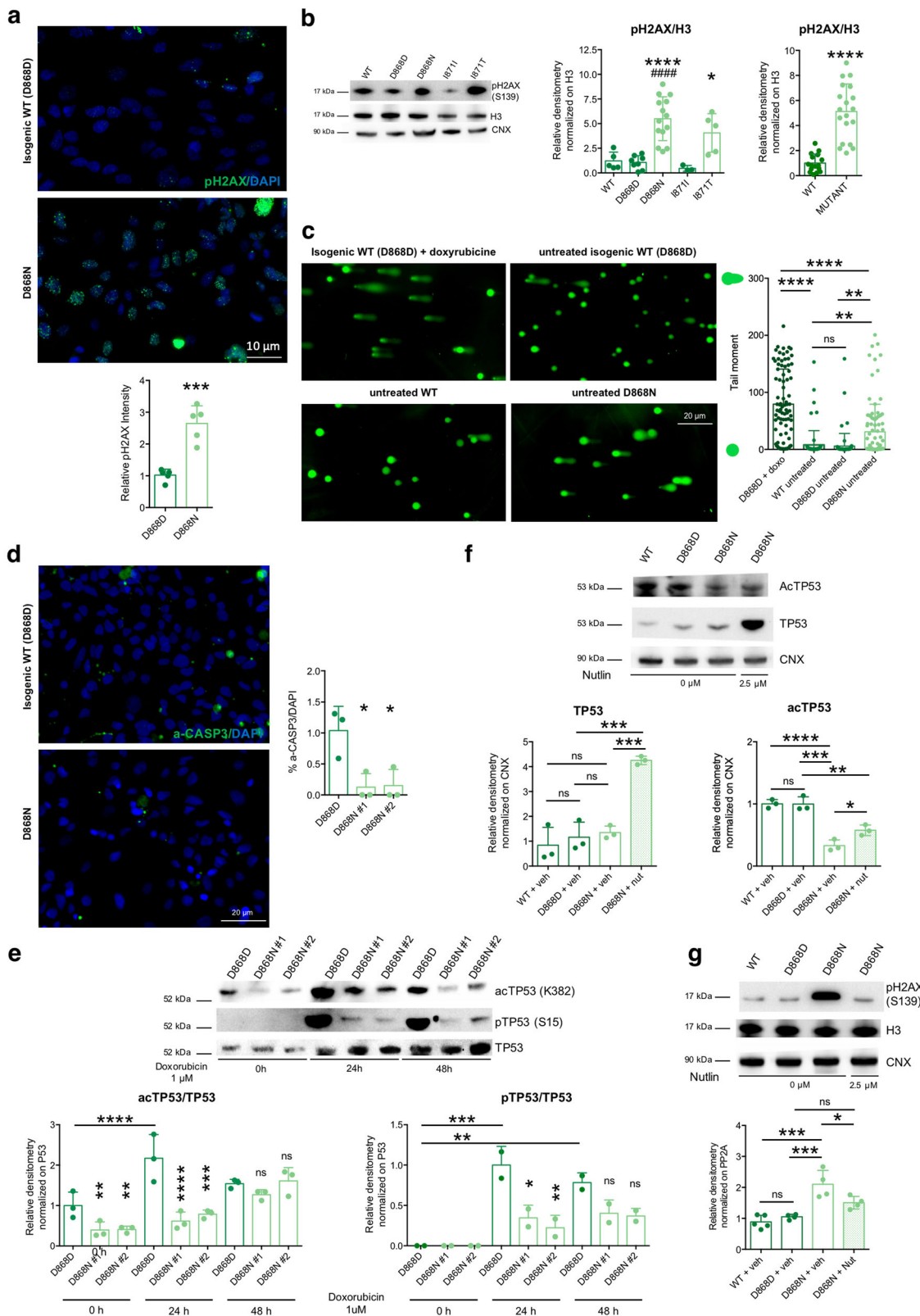

accumulated DNA damage may be the cause of genotoxic stress into intrinsically vulnerable cells as postmitotic neurons, thus possibly explaining the neurodegeneration occurring in SGS patients. To experimentally tackle this eventuality, we assessed cell death in postmitotic neuronal cultures by combining PI live staining with subsequent MAP2 immunostaining. Interestingly, dying neurons were more represented in SGS than in control cell

cultures, as detected by the increased percentage of PI-positive cells over the totality of MAP2 positive neurons (Fig. 7d). The increased cell death did not correlate with the activation of apoptotic pathways, as shown by comparable levels of both cleaved-CASPASE3 (Supplementary Fig. 10e) and cleaved-SPECTRIN (a target of active CASPASE3) in SGS and control neurons (Supplementary Fig. 10f). To exclude the possibility that the overexpression of

**Fig. 5 SETBP1 accumulation causes P53 hypofunctioning. a** Phosphorylated histone H2AX (pH2AX Ser139), immunostaining in isogenic control (D868D), and mutant (D868N) NPCs and quantification. ***$P = 0.0003$. $n = 5$. Images taken at the same magnification. **b** pH2AX (Ser139) immunoblotting and quantification in WT, isogenic control (D868D, I871I) and mutant (D868N, I871T) NPCs. * indicates statistic test between isogenic cell line pairs, # refers to the comparison between each mutant cell line with the unrelated WT one. WT vs. D868D $P = 0.9999$; WT vs. D868N $P = 0.0003$; WT vs. I871I $P = 0.9720$; WT vs. I871T $P = 0.0872$; D868D vs. D868N $P < 0.0001$; I871I vs. I871T $P = 0.0497$; D868N vs. I871T $P = 0.4922$, WT vs. MUT ****$P < 0.0001$. $n = 3$. **c** Representative images (taken at the same magnification) of alkaline comet assay in NPCs, and quantification of tail moment, in WT, isogenic control (D868D) and mutant (D868N) NPCs. D868D progenitors doxorubicin-treated (Doxo) added as a positive control. $n = 70$ cells per group from three independent experiments, D868D + Doxo vs. WT-Untreated ****$P < 0.0001$; D868D + Doxo vs. D868D-Untreated ****$P < 0.0001$; D868D + Doxo vs. D868N-Untreated ****$P < 0.0001$; WT-Untreated vs. D868D-Untreated $P = 0.9918$; WT-Untreated vs. D868N-Untreated **$P = 0.0047$; D868D-Untreated vs. D868N-Untreated **$P = 0.0015$. **d** Activated-CASPASE3 (Asp175-cleaved) immunostaining and quantification in isogenic control (D868D) and mutant (D868N) NPCs. $P = 0.0155$. $n = 3$. Images taken at the same magnification. **e** Acetylated and phosphorylated TP53 (acTP53 Lys382, pTP53 Ser15) immunoblotting upon DNA damage induction (1 µM doxorubicin treatment for the indicated time) in isogenic control (D868D) and mutant (D868N) NPCs. See Supplementary Dataset 3 for details of statistical analysis. $n = 3$. **f** l TP53 immunoblotting and quantification after Nutlin-3a treatment (WT + vehicle vs. D868D + vehicle $P = 0.8558$, D868D + vehicle vs. D868N + vehicle $P = 0.9658$, WT + vehicle vs. D868N + Nut ***$P = 0.0001$, D868N + vehicle vs. D868N + nutlin ***$P = 0.0004$, D868D + vehicle vs. D868N + nutlin ***$P = 0.0003$). acTP53 immunoblotting and quantification after Nutlin-3a treatment (WT + vehicle vs. D868D + vehicle $P = 0.9999$, D868D + vehicle vs. D868N + vehicle ***$P = 0.0001$, WT + vehicle vs. D868N + Nut **$P = 0.0022$, D868N + vehicle vs. D868N + nutlin *$P = 0.0466$, D868D + vehicle vs. D868N + nutlin **$P = 0.0024$). $n = 4$. **g** pH2AX immunoblotting and quantification after Nutlin-3a treatment (WT + vehicle vs. D868D + vehicle $P = 0.9479$, D868D + vehicle vs. D868N + vehicle ***$P = 00006$, WT + vehicle vs. D868N + Nut $P = 0.0504$, D868N + vehicle vs. D868N + nutlin *$P = 0.0354$, D868D + vehicle vs. D868N + nutlin $P = 0.1245$). $n = 4$. All data expressed as mean ± SEM. ns: nonsignificant differences when $P > 0.05$; *$P < 0.05$, **$P < 0.01$, ***$P < 0.001$, ****$P < 0.0001$. **b–g** One-way ANOVA followed by Tukey's post hoc test for multiple comparisons, except for WT vs. MUT comparisons two-sided unpaired $t$ test in **b. a** Two-sided unpaired $t$ test.

NEUROGENIN 2 may contribute to this phenotype, we generated neuronal cultures without the forced expression of the pro-neural factor using in a chemically defined medium[39]. Using this approach, we obtained equivalent results, suggesting that the DNA damage accumulation and the neurodegeneration are independent by NEUROGENIN 2 usage (Supplementary Fig. 10g, h). To investigate better the cell specificity of the observed cell death, we analyzed TBR1+ neurons, a specific subpopulation of cortical excitatory neurons of deep layers, and GABAergic inhibitory interneurons in these cultures. We could thus confirm a similar survival impairment in both subpopulations in vitro (Supplementary Fig. 10g, h). Moreover, we also differentiated into VIMENTIN+, and GFAP+ astrocytes that were obtained in a comparable number although SGS astrocytes appeared larger in size with respect to isogenic control cells (Supplementary Fig. 11a, b). SGS astrocytes displayed excessive DNA damage but, strikingly, did not show cell death at least in our culture conditions (Supplementary Fig. 11b), suggesting that the degeneration is strictly restricted to the neuronal compartment.

DNA breaks and damage activates Poly(ADP-ribose) polymerase-1 (PARP-1) which results in the transformation of nicotinamide adenine dinucleotide (NAD+) into long Poly(ADP-ribose) (PAR) polymers[40]. PAR molecules are toxic for the cell as they can be transferred to a variety of proteins and cause the release of apoptosis-inducing factors (AIF) from mitochondria, ultimately leading to a form of caspase-independent cell death referred to as parthanatos. This mechanism of cell death has been already associated with neurological pathologies, among which are neurodegenerative diseases[40–43]. We investigated the presence of PAR polymers in neurons and found a massive accumulation of PARs in mutant cells (Supplementary Fig. 12a). Of note, altered PARP-1 levels were already detectable in both SGS NPCs and organoids, but not in SGS iPSCs (Supplementary Fig. 12b). Indeed, pH2AX was not significantly increased at the pluripotent stage (Supplementary Fig. 12c). Importantly, the Nutlin-3a treatment that increased P53 levels in NPCs was also able to decrease PAR levels in SGS NPCs (Supplementary Fig. 12d).

Then, we generated cortical neurons by expressing NEUROGENIN 2 directly in undifferentiated iPSCs, thus skipping in such a way the intermediate step of proliferating NPCs (Supplementary Fig. 12d). Direct neuronal generation from iPSCs produced, MAP2+ neurons from both control and SGS genotypes with comparable efficiency (Supplementary Fig. 12d). These cells did not present differences neither in total DNA damage levels nor in PAR accumulation, while they retained both SETBP1 and SET accumulation (Fig. 7e and Supplementary Fig. 12e–i). Finally, in sharp contrast with NPC-derived neurons, neuronal cells directly originated from iPSCs did not show excessive cell death (Fig. 7f).

Next, we wondered whether inhibition of PARP-1 may reduce neuronal death in SGS. For this reason, we added the PARP-1 inhibitor Olaparib at the iPSC stage, during NPC derivation and throughout the differentiation into postmitotic neurons (Fig. 8a). This chronic PARP-1 inhibition was able to effectively reduce PAR accumulation in both SGS NPCs and neurons (Fig. 8b) and induced a mild decrease in the DNA damage marker pH2AX in both cell types (Supplementary Fig. 13a). Then, we asked whether the reduction of PARP-1 could reduce the rate of cell death in SGS neurons. Using PI/MAP2 double staining, we noticed a partial but significant decrease of neuronal cell death within SGS cultures upon Olaparib treatment (Fig. 8c). Of note, the usage of either the broad-spectrum caspase inhibitor carbobenzoxy-valyl-alanyl-aspartyl-[O-methyl]-fluoromethylketone (Z-VAD) or the necroptosis inhibitor Nec-1s had no appreciable effects (Fig. 8c). As mentioned above, PARP-1 uses NAD+ to produce PARs[44]. NAD+ is a critical co-factor for cellular energy production, and its excessive consumption might lead to metabolic failure and consequent cell death[45]. As expected, SGS neurons showed lower NAD+/NADH content when compared to their isogenic counterparts (Fig. 8d). Importantly, the supplement of exogenous NAD+ could partially rescue cell death in SGS neurons (Fig. 8e).

The very high level of PARP-1 activation in NPCs did not lead to cell death, indicating that the parthanatos was anyhow prevented at this stage. However, the strong PARylation we observed (Supplementary Fig. 12b) may have other toxic effects beyond parthanatos and might contribute to the transformed identity and the pathological traits of SGS NPCs. Recent findings disclosed an alternative mechanism of PARP-1 function that involves the direct interaction with phosphorylated ERK2 (externally regulated kinase), resulting in an amplification of downstream signals that mediates expression of immediate early genes, such as the members of the FOS and JUN gene families[46]. The latter has been implicated in promoting cell proliferation, both in normal and in transformed cells through multiple

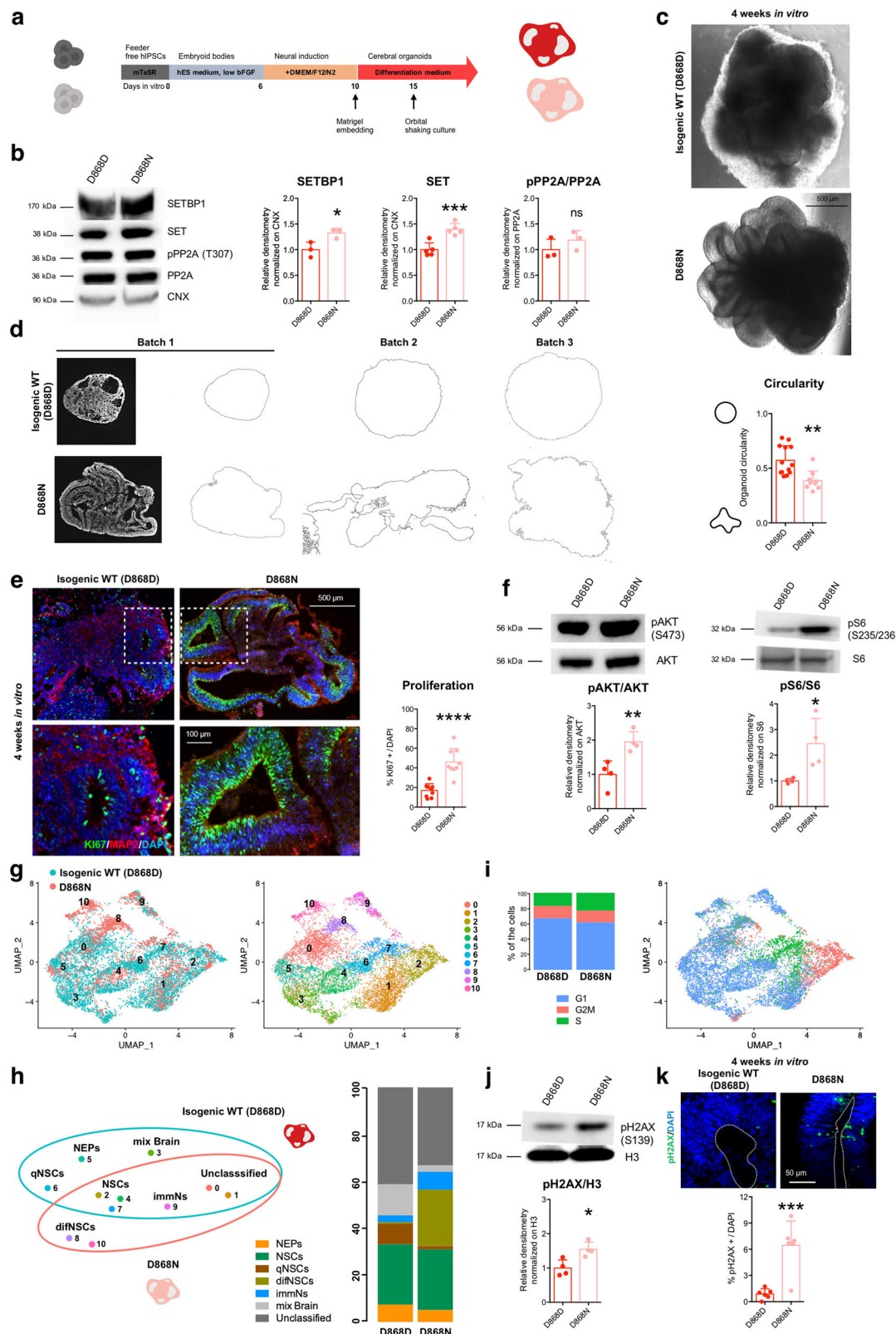

mechanisms, including regulation of CYCLIN D1, E2F factors, and their target genes[47–49]. SGS NPCs displayed a very high expression of these genes (Fig. 4 and Supplementary Fig. 4) that were blunted by Olaparib (Fig. 8f). Accordingly, the decrease of PARP-1 activation obtained by Olaparib administration led to a normalization of the abnormal cell proliferation in SGS NPCs (Fig. 8g, h).

Altogether, these results suggest that the heightened DNA damage accumulated in SGS NPCs and inherited by their neuronal derivatives is toxic, at least in part because of the underlying PARP-1 overactivation and subsequent PARs accumulation. This leads to both the self-sustaining overproliferative behavior of SGS neural precursors and the degeneration through parthanatos in the derived neurons. Thus, our data indicate that

**Fig. 6 Mutant organoids recapitulate SGS-associated features. a** Scheme for cerebral organoid generation protocol. **b** SETBP1-SET-PP2A axis in cerebral organoids: immunoblotting for SETBP1 ($*P = 0.0329$, $n = 3$), SET ($***P = 0.0009$, $n = 5$) and pPP2A/PP2A ratio ($P = 0.3084$, $n = 3$) in isogenic control (D868D) and mutant (D868N) organoids. **c** Representative bright-field images (taken at the same magnification) of D868D and D868N organoids. **d** Evaluation of gross morphological appearance of control and mutant organoids. Images (taken at the same magnification) of serial histological sections from 4-week-old organoids from three independent batches were converted into outlines for the quantification of the perimeter and circularity. $**P = 0.0015$. b = 3 (batches), o ≥ 9 (organoids). **e** KI67 and MAP2 immunostaining of cryo-sections of D868D and D868N organoids. Insets indicate the magnifications below (left). Quantification of KI67 + /DAPI cells in representative ventricular-like structures (right, $****P < 0.0001$). b = 3 (batches), o = 3 (organoids), v = 9 (ventricles). Images taken at the same magnification. **f** Left, total and phosphorylated AKT (pAKT Ser473) immunoblotting and quantification of pAKT/AKT ratio in D868D and D868N organoids. ($**P = 0.0085$). Right, total and phosphorylated S6 (pS6 pS6 Ser235, pS6 Ser236) immunoblotting and quantification of pS6/S6 ratio in D868D and D868N organoids. ($*P = 0.0253$,). $n = 4$. **g** Left, UMAP plot showing the distribution of cells from 4-week-old organoids of SGS genotype (D868N, salmon) and isogenic control (D868D, light blue). Right, UMAP plot of cells from 4-week-old organoids of D868N and D868D showing 10 clusters (resolution 0.7). **h** Left, cluster categorization (NEPs = neuroepithelial cells; NSCs = neural stem cells; qNSCs = quiescent NSCs; difNSCs = differentiating NSCs; immNs = immature neurons; mix Brain = cells of several stages from different cell regions) and diffusion among genotypes. Right, bar plots of the enrichment of the different cluster categories within the two genotypes. **i** Left, cell cycle distribution in 4-week-old organoids of D868N and D868D. Right, UMAP plot of cells from 4-week-old organoids of D868N D868D showing cell cycle distribution. **j** pH2AX (Ser139) immunoblotting and quantification in D868D and D868N organoids. ($**P = 00019 n = 4$). **k** Phosphorylated histone H2AX (pH2AX Ser139, immunostaining in rosettes of isogenic control and mutant 4-week-old organoids ($***P = 0.0008$). A dashed line indicates ventricular surface. b = 3 (batches), o = 3 (organoids), v = 6 (ventricles). Images taken at the same magnification. All data expressed as mean ± SEM. ns: nonsignificant differences when $P > 0.05$; $*P < 0.05$, $**P < 0.01$, $***P < 0.001$, $****P < 0.0001$. **b, d, e, f, j, k** Data analyzed by two-sided unpaired $t$ test.

SGS early neurodegeneration stems in the cancer-related alterations developed in patient neural progenitors, which will give rise to a population of already intoxicated neurons prone to premature cell death.

## Discussion

In this study, we revealed that *SETBP1* mutations, known to be responsible for Schinzel–Giedion syndrome, lead to the accumulation of the SETBP1 and SET proteins and the consequent P53 inhibition in neural cells. These molecular changes promote the onset of cancer-like behavior in neural progenitors that accumulate widespread DNA damage without programmed cell death engagement. Repetitive genomic breaks stimulate PARP-1 activity generating excessive PARs. This exacerbates the overproliferation of NPCs on one side, and on the other the cell death of the descendent postmitotic neurons. This direct causative relationship between SETBP1/SET/P53, cancer-like behavior, and subsequent neurodegeneration shed light on the pathological mechanisms at the basis of the severe neurological pathology in SGS (Supplementary Fig. 13b).

We first showed that SGS mutations cause an accumulation of SETBP1 protein, confirming in neural cells the proposed model for the mutations clustered in the degron sequence[12,13]. Interestingly, SGS cells do not show any sign of protein accumulation at the pluripotent stem cell stage. It is plausible that the intrinsic characteristics of this early cell type could mask this effect and buffer this accumulation. In fact, these embryonic cells developed different mechanisms to cope with protein clearance in respect to more differentiated cells, thus ensuring a tight proteostasis control, which is, in turn, essential for their high self-renewal rate, pluripotency and cell fate decisions[50–52]. For instance, ESCs and iPSCs properties depend on enhanced global translational rates and upregulation of both the autophagic and proteasomic activities[53,54]. In particular, in pluripotent stem cells, the ubiquitin–proteasome system (UPS) is relying on multiple and redundant players that confer the ability to recognize multiple consensus target sequences[55]. Therefore, iPSCs have multiple pathways to stimulate protein clearance potentially impeding the accumulation of the mutated SETBP1 protein.

On the other hand, SETBP1 accumulation in both NPCs and neurons promotes SET stabilization without detectable alterations in PP2A activity, which has been proposed as one oncogenic trigger in leukemia[12,13]. SGS NPCs developed a cancer-like behavior, both phenotypically—dramatic increase of proliferation, lack of apoptosis—and molecularly—upregulation of oncogenes and downregulation of oncosuppressors, including the deregulation of MAPK, WNT, NOTCH, and PI3K/AKT pathways. These changes are in line with the clinical observations in SGS patients who are described with (i) large ventricles at birth that can represent a sign of tangential expansion due to overproliferation within the ventricular zone of the cerebral cortex[7,10,56] and (ii) a high incidence of solid tumors of neuroepithelial origin[7].

Recently, it has been suggested that the phosphoinositide-dependent kinase 1 (PDK1)-AKT axis also plays a role in regulating neuronal migration during embryogenesis[57]. Indeed, this pathway controls many key mechanisms required for cell migration, including microtubule stabilization and cell polarization[58]. Thus, it is not surprising that PI3K activity is required for normal radial migration in the developing cerebral cortex[59]. Interestingly, we showed that acute overexpression of SETBP1 during active neurogenesis in mouse induces delayed migration of postmitotic neurons toward their final seat in the forming layers of the neocortex[22]. This defect may also play a role in SGS patients, determining peculiar cortical organization with possible layering defects. Further investigations, including human studies and the employment of animal models, will clarify this possibility.

Our data strongly suggest that deficiency in P53 signaling caused by direct SET inhibition plays a central role in SGS pathogenesis[32]. It would be interesting to evaluate whether this mechanism may contribute to the oncogenic transformation in leukemic patients carrying *SETBP1* mutations. Future studies will determine if SETBP1 accumulation might also have SET-independent functions; for instance, by regulating epigenetic players with whom it was shown to directly interact[22].

Alterations in developmental processes are a common cause of cancer, with several genes and molecular pathways implicated in both developmental diseases and cancer in humans[60]. *SETBP1* belongs to a group of genes where identical point mutations can lead to cancer, when mutated somatically, or to severe developmental syndromes when occurring in the germline. Beyond *SETBP1* this group includes *ASXL1* (myeloid malignancies and Bohring–Opitz syndrome), *EZH2* (B-cell lymphoma and Weaver syndrome) as well as several genes of the MAP kinase pathway, *FGFR2, FGFR3, HRAS, PTPN11, BRAF, MAP2K1* (many types of cancers and paternal age–effect disorders such as Noonan syndrome and others)[61]. A pleiotropic mechanism might predict an

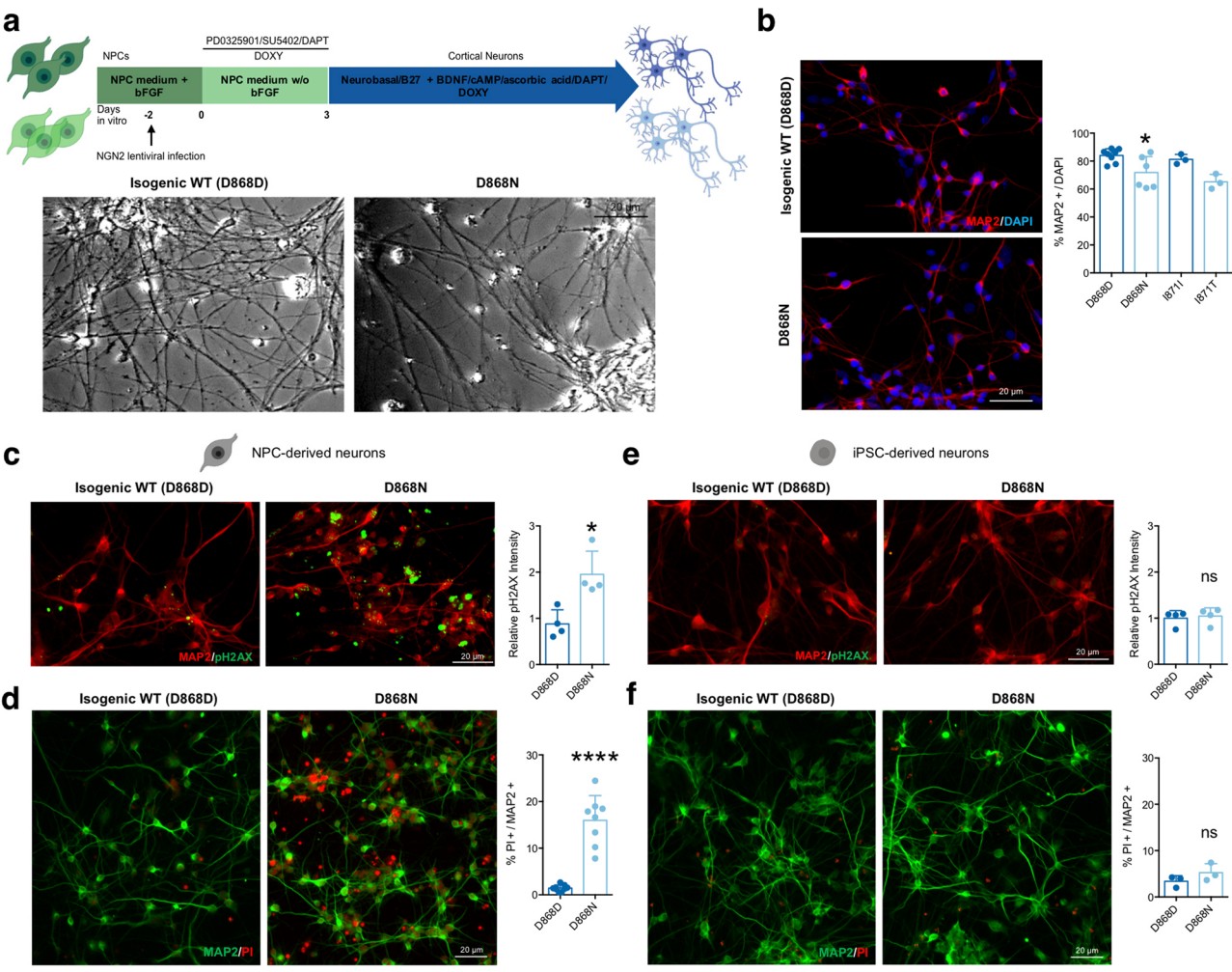

**Fig. 7 SGS neurons inherit DNA damages and are prone to degenerate. a** Scheme for NPC-derived neuronal differentiation protocol (upper panel) and representative bright-field images (taken at the same magnification) of isogenic control (D868D) and mutant (D868N) neuronal cultures (lower panel). **b** Representative images (taken at the same magnification) of immunostaining for MAP2 (left) and relative quantification of isogenic control (D868D) and mutant (D868N) NPC-derived neuronal cultures. D868D vs. D868N *P = 0.0370; I871I vs. I871T P = 0.0768; D868N vs. I871T P = 0.5995. One-way ANOVA followed by Tukey post hoc test for multiple comparisons. n = 3. **c** Immunostaining and quantification for phosphorylated histone H2AX (pH2AX Ser139) in isogenic control (D868D) and mutant (D868N) NPC-derived neuronal cells. *P = 0.0107. Two-sided unpaired t test. n = 4. Images taken at the same magnification. **d** Representative images (taken at the same magnification) of immunostaining for MAP2 and PI and relative quantification of PI⁺ over MAP2⁺ cells in isogenic control and mutant NPC-derived neurons. ****P < 0.0001. Two-sided unpaired t test. n = 8. **e** Immunostaining and quantification for phosphorylated histone H2AX (pH2AX Ser139) in isogenic control (D868D) and mutant (D868N) iPSC-derived neuronal cells. P = 0.7200. Two-sided unpaired t test. n = 4. Images taken at the same magnification. **f** Representative images of immunostaining for MAP2 and PI and relative quantification of PI⁺ over MAP2⁺ cells in isogenic control and mutant iPSC-derived neurons. P = 0.2337. Two-sided unpaired t test. n = 3. Images taken at the same magnification. All data expressed as mean ± SEM. ns: nonsignificant differences when P > 0.05; *P < 0.05, ****P < 0.0001. Analysis was performed at 4 weeks of neuronal differentiation.

independent origin of the developmental defects and cancer by assuming that these genes have unrelated functions over time and in different genetic backgrounds and cellular contexts. Alternatively, this pathological convergence may indicate that common alterations in central cellular processes such as chromatin remodeling, cell proliferation, and epithelial–mesenchymal transitions might profoundly alter embryogenesis and promote cancer development. In this respect, SETBP1 represents a hub for all these functions, having been associated with cell proliferation, epigenetics, and strong upregulation of MAPK signaling.

In this scenario, tumor-like SGS NPCs accumulate excessive DNA damage that is directly inherited by the descendent neurons. Accumulation of genomic insults may result from either the increased damaging conditions or ineffective DDR, or both. These alterations might co-occur in SGS NPCs, where a sustained

rate of cell proliferation that promotes genotoxicity per se is present together with blunted DNA damage repair mechanisms, due to P53 loss of activity. Interestingly, other P53-independent SET functions might contribute to ineffective DNA repair. In fact, SET was shown to regulate the Ku70/80-mediated non-homologous-end-joining (NHEJ) repair by inhibiting CBP acetylation of Ku70[62] and to promote chromatin compaction to negatively regulate repair by homologous recombination in cancer cells[63].

The integrity of DNA in postmitotic cells like neurons is crucial for their functions and survival. DNA damage accumulation can give rise to deleterious instability leading to cell senescence or cell death occurring in aging or age-related diseases, including Parkinson's disease[64–66]. Moreover, neurological dysfunctions and neurodegeneration are distinguished hallmarks in diseases caused

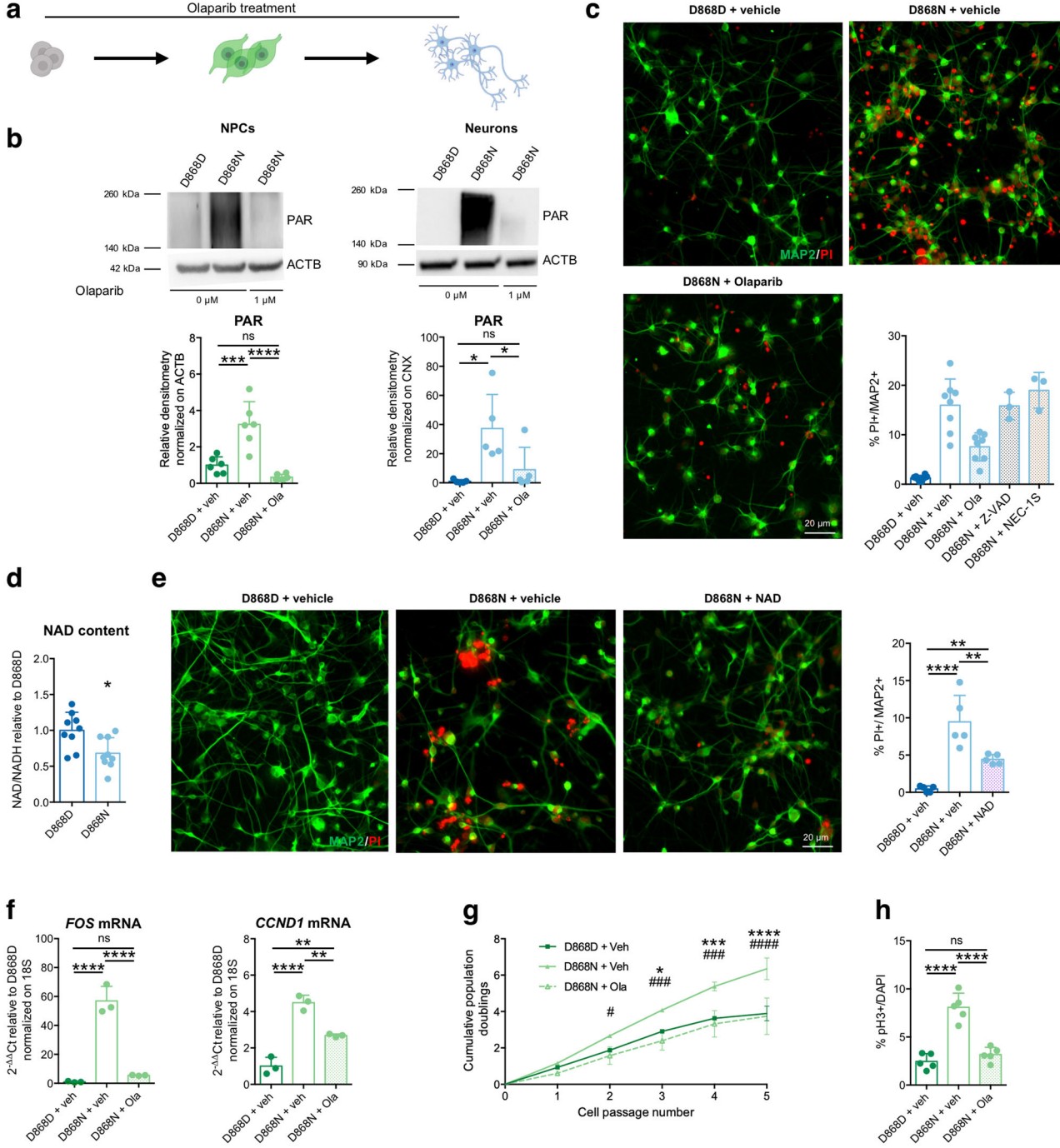

by mutations in DNA repair genes[65,67]. In particular, *ATM* gene mutations are the most frequent cause of ataxia telangiectasia, an autosomal recessive disorder characterized by cerebellar degeneration[68]. Also, genetic syndromes due to DNA repair impairment such as xeroderma pigmentosum, Cockayne syndrome, and trichothiodystrophy, develop both neurodegeneration and an elevated predisposition to tumors, closely resembling SGS[65].

In this study, we identified parthanatos, a cell death program mediated by PAR accumulation, as a leading cause of SGS neurodegeneration[40,41]. Mechanistically, unsolved DNA damage stimulates PARP-1 activation predisposing neurons to premature cell death. PARP-1 has been already associated with neurodegeneration in Parkinson's disease where PARs are the key mediators of α-synuclein-associated neurodegeneration[43] and

cerebellar ataxia with mutations in the *XRCC1* gene, which has a fundamental role in single-strand break repair[69]. Recent work also demonstrated that deletion of Topoisomerase 1 in postmitotic excitatory neurons causes genomic instability, PARP-1 overactivation, and early neurodegeneration in a mouse model[70]. However, this work provides the first evidence that SETBP1 protein levels control P53 activity through the stabilization of the SET protein. Alterations of this molecular circuitry elicit DNA repair defects and damage accumulation ultimately leading to cell death by parthanatos exclusively in postmitotic neurons. In fact, high levels of PARP-1 activation in neural progenitors did not evidently alter their survival. A plausible explanation for this is that NPCs are highly proliferating cells and rely mainly on aerobic glycolysis rather than mitochondrial respiration for

**Fig. 8 Rescue of PARP-1 activation. a** Olaparib treatment of mutant (D868N) cells. D868D and D868N cells were exposed to vehicle (DMSO) with the same protocol. **b** PAR polymers immunoblotting and quantification in NPCs (left; D868D + vehicle vs. D868N + vehicle ***$P = 0.0004$; D868D + vehicle vs. D868N + Olaparib $P = 0.3262$; D868N + vehicle vs. D868N + Olaparib ****$P < 0.0001$, $n = 6$) and NPC-derived neurons (right; D868D + vehicle vs. D868N + vehicle *$P = 0.0106$; D868D + vehicle vs. D868N + Olaparib $P = 0.7230$; D868N + vehicle vs. D868N + Olaparib *$P = 0.0425$, $n = 5$). **c** Representative images (taken at the same magnification) of MAP2 and PI immunostaining of mock-treated isogenic control (D868D), mock-treated mutant (D868N), and Olaparib-treated mutant (D868N)-NPC-derived neurons. Quantification of PI+/MAP2+ cells in mock-treated isogenic control ($n = 8$)-, mock-treated mutant ($n = 8$)-, Olaparib-treated mutant ($n = 8$)-, Z-VAD-treated mutant ($n = 3$)-and NEC-1S-treated mutant ($n = 3$)-NPC-derived neuronal cultures. See Supplementary Dataset 3 for statistical analysis. $n = 3$ differentiation experiments. **d** Quantification of NAD+/NADH content in D868DD868N NPC-derived neurons. *$P = 0.0108$. Two-sided unpaired $t$ test. $n = 9$ wells over three differentiation experiments. **e** Representative images (taken at the same magnification) of MAP2 and PI immunostaining and quantification of PI+/MAP2+ cells in mock-treated isogenic control (D868D)-, mock-treated mutant (D868N)-, and NAD+-supplemented mutant (D868N) NPC-derived neurons. D868D + vehicle vs. D868N + vehicle ****$P < 0.0001$; D868D + vehicle vs. D868N + NAD+ *$P = 0.0277$; D868N + vehicle vs. D868N + NAD+ **$P = 0.0068$. $n = 5$. **f** RT-qPCR analysis for *FOS* and *CCND1* mRNA restored expression levels after Olaparib treatment in mutant NPCs. *FOS*: D868D + vehicle vs. D868N + vehicle ****$P < 0.0001$; D868D + vehicle vs. D868N + Olaparib $P = 0.6371$; D868N + vehicle vs. D868N + Olaparib ****$P < 0.0001$. *CCND1*: D868D + vehicle vs. D868N + vehicle ****$P < 0.0001$; D868D + vehicle vs. D868N + Olaparib **$P = 0.0034$; D868N + vehicle vs. D868N + Olaparib **$P = 0.0024$. $n = 3$. **g** Growth curve analysis of mock-treated isogenic control (D868D)-, mock-treated mutant (D868N)-, and Olaparib-treated mutant (D868N)-NPCs. "*" indicates statistical analysis between "D868N + vehicle" and "D868D + vehicle", while "#" indicates statistical analysis between "D868N + vehicle" and "D868N + Olaparib". Two-way ANOVA followed by Tukey post hoc test for multiple comparisons. See Supplementary Dataset 3 for details of statistical analysis. $n = 3$. **h** Quantification of pH3 immunostaining in isogenic control (D868D)-mock-treated, mutant (D868N)-mock-treated, and mutant (D868N)-Olaparib-treated NPCs. D868D + vehicle vs. D868N + vehicle ****$P < 0.0001$; D868D + vehicle vs. D868N + Olaparib $P = 0.5530$; D868N + vehicle vs. D868N + Olaparib ****$P < 0.0001$. $n = 5$. All data expressed as mean ± SEM. ns: nonsignificant differences when $P > 0.05$; *$P < 0.05$. **$P < 0.01$. ***$P < 0.001$. ****$P < 0.0001$. **b, c, e, f, h** One-way ANOVA followed by Tukey post hoc test for multiple comparisons. Analysis was performed at 4 weeks of differentiation for neuronal cultures.

energy supply[71], and therefore are less sensitive to NAD+ deprivation.

SGS patients suffer from intractable epilepsy that highly contributes to their poor quality of life. The origin of the seizures is currently unknown. One possibility is that the neurodegeneration occurring throughout the cerebral cortex of the patients may lead to unbalanced network activity driving the insurgence of the attacks. Intense PARP-1 activation, through NAD depletion and reduction of SIRT1 deacetylase activity, was indicated as the main cause of neuronal death in an in vitro model of acute acquired epilepsy[72]. Importantly, mutations in the hydrolases ARH3 and TARG, the enzymes involved in removing ADP-ribose modifications, also lead to neurodegenerative diseases presenting epileptic manifestations[73–75]. Thus, if excessive PARP-1 activity, PAR levels, NAD+ depletion, and/or parthanatos are responsible for neurological defects seen in SGS patients, either currently available catalytic PARP-1 inhibitors or NAD+ repletion, may represent valuable therapeutic options[76]. However, further studies are necessary to clarify whether such treatments will be effective in postmitotic neurons that already inherited damage accumulation from the mitotic precursor stages.

Our data revealed that PARP-1 also contributes to the high proliferation rate of SGS NPCs by activation of the FOS/JUN genes. Several members of these families can heterodimerize, forming different activating protein-1 (AP-1) transcription factors. AP-1 inhibition has been shown as beneficial in several cancer cell lines by decreasing their growth potential and thus been identified as a potential target for cancer prevention and therapy in many types of tumors[77–82]. PARP-1 inhibition represents a current strategy for cancer treatment to prevent heightened AP-1 activation[46].

In conclusion, we report the development of a human in vitro system to model SGS pathological traits and test candidate therapeutic compounds. This model enabled us to unveil a direct relationship between the accumulation of SETBP1 and SET proteins and P53 signaling inhibition. These alterations promote a cancer-like behavior in SGS NPCs with increased proliferation and DNA damage accumulation. Genotoxic stress leads to massive activation of PARP-1 which triggers neuronal cell death, explaining the early neurodegeneration observed in SGS patients.

This model warrants further investigations in other genetic forms of neurodegenerative disorders, in which the cues for neurodegeneration may possibly have originated during the development of the brain. Intriguingly, our findings indicate that pharmacological PARP-1 inhibition and NAD+ boosting compounds hold promise as novel therapeutic agents for SGS, which to date has no cure. More studies on SGS will reveal mechanisms at the crossroad between oncogenesis, brain development, and neurodegeneration that likely will be shared with other more common pathological conditions.

## Methods

**Fibroblast culture.** The patient fibroblasts harboring heterozygous mutations I871T and D868N in the *SETBP1* gene and aged-matched control fibroblasts were obtained from Alexander Hoischen (Radboud University Medical Center, Nijmegen, The Netherlands). The protocol was approved by the Medical Ethics Committee of the Radboud University Medical Center and written consent to participate was obtained for all patients. All human skin fibroblast (HSF) cultures were maintained in DMEM medium (Dulbecco's Modified Eagle's Medium—high glucose, Sigma-Aldrich) containing 10% fetal bovine serum (FBS, Sigma-Aldrich), 1% Pen/Strept, 2 mM glutamine (Sigma-Aldrich), 1% nonessential amino acids (MEM NEAA, ThermoFisher Scientific), 1% sodium pyruvate solution (Sigma-Aldrich). The cultures were kept in a humidified atmosphere of 5% $CO_2$ at 37 °C under atmospheric oxygen conditions.

**Fibroblast reprogramming and iPSC culture.** Human skin fibroblasts (HSFs) were reprogrammed to pluripotency using CytoTune®-iPS Reprogramming Kit (Invitrogen) following the manufacturer's instructions. iPSC cell lines were maintained in feeder-free conditions in mTeSR1 (Stem Cell Technologies) supplemented with Pen/Strept (1%, Sigma-Aldrich) and seeded on human embryonic stem cell (HESC)-qualified Matrigel (Corning)-coated six-well plates; cells were fed daily and passaged in cell clumps weekly using Accutase solution (Sigma-Aldrich). iPSCs were used at passages between 25 and 50 for all the subsequent experiments.

**CRISPR-Cas9 iPSC editing.** sgRNAs were designed using optimized CRISPOR (http://crispor.tefor.net/) to screen for highly selective sgRNAs in the region of interest surrounding patients' mutations. Then, selected sgRNAs were cloned using BsmBI restriction enzyme in the U6-filler-sgRNA scaffold cassette derived from LentiCRISPR v2 (Addgene 52961). This cassette was previously subcloned, as described by Giannelli et al.[83], in a lentiviral backbone carrying blasticidin antibiotic resistance under the control of Ef1a core promoter.

Wild-type versions of sgRNAs were tested in HEK293T (ATCC, Cat# CRL-3216); expression plasmids carrying different sgRNAs were co-transfect with expression plasmid pCAG-Cas9-P2A-PuroR[83] using Lipofectamine LTX Reagent (ThermoFisher Scientific); 24 h after cell transfection, a double-antibiotic selection

was applied (Blasticidin 10 μg/ml, ThermoFisher Scientific and puromycin 1 μg/ml, Sigma-Aldrich) until complete death of non-transfected cells (48 h). Surviving cells for each tested sgRNA were collected to obtain genomic DNA to perform a PCR using specific primers (fw: 5'-GAAGGCGTGTGTGAATGTGT-3'; rev: 5'-CAGTAAGGCTTCCGGTCAAC-3') on the sgRNA target region and their efficiency was calculated performing the T7 Endonuclease I (T7EI).

Mutant versions of selected sgRNAs were cloned in the previously described expression plasmid. iPSCs were co-transfected using Lipofectamine Stem Transfection Reagent (ThermoFisher Scientific) with plasmids carrying mutant version patient-specific sgRNAs and pCAG-Cas9-P2A-PuroR and a 120 bps ssODN as a homologous repair template. After a double-antibiotic selection performed as previously described, surviving cells underwent a limiting dilution step to obtain the subsequent formation of single-cell clones, which were then picked and amplified till they reach a sufficient quantity to obtain genomic DNA to perform a PCR. To verify mutations' correction, purified (Wizard SV gel and PCR Clean-Up System, Promega) PCR products were subjected to Sanger sequencing (GATC Biotech) after the usage of Zero Blunt Topo PCR cloning kit (ThermoFisher Scientific), to clone PCR product of single allele from each iPSCs clone. Genotyping of parental original iPSC lines was performed with the same procedure.

*Genomic DNA extraction.* Genomic DNA was extracted by incubating cells for 4–24 h at 55 °C in a digestion buffer mix (100 mM Tris-HCl pH8, 200 mM NaCl, 5 mM EDTA, 0.5% SDS with proteinase K, Sigma). Then, genomic DNA was purified using isopropanol, washed with ethanol 70%, and resuspended in water.

**T7 Endonuclease I (T7EI) assay.** For T7EI analysis, 200 ng of purified PCR products (Wizard SV gel and PCR Clean-Up System, Promega) were denatured and reannealed in NEB buffer 2 (New England Biolabs). We used the following protocol: 95 °C, 5 min; 95–85 °C at − 2 °C/s; 85–25 °C at − 0.1 °C/s; hold at 4 °C. Then, 10 U of T7EI (New England Biolabs) was added and an incubation step at 37 °C for 15 min was performed. PCR products were analyzed on 2% agarose gels and imaged with a Gel Doc gel imaging system (Bio-Rad). The percentage of genome modification was obtained as previously described[84].

Briefly, estimation of the cleavage intensity was obtained by measuring the integrated intensity of the PCR amplicon and cleaved bands by using Fiji software. For each lane, the fraction of the PCR product cleaved (fcut) was calculated by using the following formula: $f_{cut} = (b + c)/(a + b + c)$, where a is the integrated intensity of the undigested PCR product and b and c are the integrated intensities of each cleavage product. Indel occurrence can be estimated with the following formula, based on the binomial probability distribution of duplex formation: indel (%) $= 100 \times (1−\sqrt{(1−f_{cut})})$.

*Off-target analysis.* The potential off-target sites of both sgD868N_6 and sgI871T_9 were selected according to online tool: http://crispor.tefor.net. In brief, we chose the top five most likely off-targets hits for each sgRNAs falling in gene-coding regions. These sites were amplified by genomic PCR (primers are listed in Supplementary Dataset 4) and obtained amplicons underwent Sanger sequencing in order to assess the absence of alterations.

**Karyotype analysis.** The cytogenetic analysis was performed on iPSC by standard procedures. Briefly, active cell division was blocked at metaphase by 50 μg/ml of colcemid (Irvine Scientific) for 2–3 h at 37 °C. Thereafter, cells were detached using trypsin–EDTA, subsequently incubated in 1% hypotonic solution (sodium citrate tribasic dihydrate) and then fixed with Carnoy's fixative (3:1 methanol to acetic acid) onto glass slides. Q-banded metaphases were analyzed and interpreted according to the International System for Human Cytogenetic Nomenclature [ISCN 2013]. A minimum of 16 metaphase spreads per sample were analyzed with an Olympus BX51 microscope coupled to a charge-coupled device camera COHU 4912 (Olympus, Milan, Italy). Captured images were analyzed using Ikaros (v 5.8.12) (MetaSystems).

**Multi-germ layer differentiation.** iPSCs at 70–80% confluence were detached by Accutase solution incubation at 37 °C for 10 min in order to obtain a single-cell suspension. Cells were centrifuged, counted, and seeded onto HESC-qualified Matrigel-coated glass coverslips at a density of 26,000 cells/cm² in mTeSR1 medium supplemented with 10 μM ROCK inhibitor Y27632 (Selleckchem) and 1% Pen/Strept. Twenty-four hours after seeding, medium was replaced with DMEM/F12 containing 1% Pen/Strept, 2 mM glutamine (Sigma-Aldrich), 1% nonessential amino acids (MEM NEAA, ThermoFisher Scientific), 10% fetal bovine serum (FBS, Sigma-Aldrich). Cells were cultured for 4 weeks with medium replacement three times per week and then analyzed by immunofluorescence.

**2D neural differentiation**

*NPC generation and culture.* NPCs were generated as previously described with appropriated optimization[27]. Briefly, iPSCs were dissociated in cell clusters using Accutase solution (Sigma-Aldrich) and seeded onto low-adhesion plates in mTeSR1 supplemented with 0.5× N-2 supplement (ThermoFisher Scientific), Pen/Strept (1%, Sigma-Aldrich), human Noggin (0.5 μg/ml, PeproTech), SB-431542 (5

μM, Sigma-Aldrich) and ROCK inhibitor Y27632 (10 μM). The medium was replaced every 3 days. After 10 days, embryoid bodies were seeded onto matrigel-coated plates (1:100, Matrigel growth factor reduced, Corning) in DMEM/F12 (Sigma-Aldrich) supplemented with 1× N-2 supplement, nonessential amino acids (1%, MEM NEAA, ThermoFisher Scientific) and Pen/Strept. The medium was replaced every other day. After 10 days, rosettes were dissociated with Accutase solution and NPCs (P0) were plated onto matrigel-coated-flasks in NPC media containing DMEM/F12, 0.5× N-2, 0.5× B-27 supplement (ThermoFisher Scientific), Pen/Strept (1%), and bFGF (20 ng/ml, ThermoFisher Scientific). NPCs were passaged twice a week using Accutase solution; experiments were performed with NPCs between P3 and P10.

*NPC-derived postmitotic neurons differentiation (with NGN2 overexpression).* At differentiation day −2, 90% confluent NPC cultures were infected with the lentiviral vector TetO-Ngn2-T2A-Puro[37] in NPC medium supplemented with doxycycline (Sigma-Aldrich, 2 μg/ml), overnight. The next day, the medium was replaced with fresh NPC medium supplemented with antibiotic selection (puromycin 1 μg/ml) and doxycycline; doxycycline was maintained for all the experiments. On day 0, medium was replaced with differentiation medium; differentiation medium was composed of NPC medium without bFGF, supplemented with SU5402 (Sigma-Aldrich, 10 μM), PD0325901 (Sigma-Aldrich, 8 μM) DAPT (Sigma-Aldrich, 10 μM). Differentiation medium was replaced every day with a fresh one on days 1 and 2. On day 3, cells were detached by Accutase solution incubation at 37 °C for 20 min in order to obtain a single-cell suspension. Cells were centrifuged, counted, and seeded at a density of 55,000 cells/cm² onto Poly-L-lysine/laminin/fibronectin (all from Sigma-Aldrich, 100 μg/ml, 2 μg/ml, 2 μg/ml)-coated plates or coverslip in neuronal maturation medium supplemented with ROCK inhibitor Y27632 (10 μM) for the first 24 h. Neuronal maturation medium was composed by Neurobasal A (ThermoFisher Scientific) supplemented with 1× B-27 supplement, 2 mM glutamine, 1% Pen/Strept, BDNF (Peprotech, 20 ng/ml), ascorbic acid (Sigma-Aldrich, 100 nM), laminin (1 μg/μl), DAPT (10 μM), dbcAMP (Selleckchem, 250 μM). The culture medium was replaced the next day to remove ROCK inhibitor, and then half of the medium was replaced with a fresh neuronal maturation medium twice a week.

*NPC-derived postmitotic neurons differentiation (without NGN2 overexpression).* On day 0, medium of 90% confluent NPC cultures was replaced with differentiation medium; differentiation medium was composed of NPC medium without bFGF, supplemented with SU5402 (Sigma-Aldrich, 10 μM), PD0325901 (Sigma-Aldrich, 8 μM) DAPT (Sigma-Aldrich, 10 μM). Differentiation medium was replaced every day with a fresh one on days 1 and 2. At day 3, cells were detached by Accutase solution incubation at 37 °C for 20 min in order to obtain a single-cell suspension. Cells were centrifuged, counted, and seeded at a density of 55,000 cells/ cm² onto poly-L-lysine/laminin/fibronectin (all from Sigma-Aldrich, 100 μg/ml, 2 μg/ml, 2 μg/ml)-coated plates or coverslip in neuronal maturation medium supplemented with ROCK inhibitor Y27632 (10 μM) for the first 24 h. Neuronal maturation medium was composed by Neurobasal A (ThermoFisher Scientific) supplemented with 1× B-27 supplement, 2 mM glutamine, 1% Pen/Strept, BDNF (Peprotech, 20 ng/ml), ascorbic acid (Sigma-Aldrich, 100 nM), Laminin (1 μg/μl), DAPT (10 μM), dbcAMP (Selleckchem, 250 μM). The culture medium was replaced the next day to remove the ROCK inhibitor, and then half of the medium was replaced with a fresh neuronal maturation medium twice a week.

*iPSC-derived postmitotic neurons differentiation.* At differentiation day −2, 90% confluent iPSC cultures were infected with the lentiviral vector TetO-Ngn2-T2A-Puro[37] in mTeSR1 medium supplemented with doxycycline (2 μg/ml), overnight. The next day, the medium was replaced with fresh mTeSR1 medium supplemented with antibiotic selection (puromycin 1 μg/ml) and doxycycline; doxycycline was maintained for all the experiments. At day 0, medium was replaced with differentiation medium "mTeSR1 + LSBX". Differentiation medium was replaced daily according to the following scheme:

Day 0, 1 : mTeSR1 + LSBX

Days 2, 3: mTeSR1 + LSBX + PSD

Days 4, 5:2/3mTeSR1 + 1/3N − 2 medium + LSX + PSD

Days 6, 7:1/3mTeSR1 + 2/3N − 2 medium + PSD

At day 8, cells were detached by Accutase solution incubation at 37 °C for 20 min in order to obtain a single-cell suspension. Cells were centrifuged, counted, and seeded at a density of 55,000 cells/cm² onto poly-L-lysine/laminin/fibronectin-coated plates or coverslip in neuronal maturation medium supplemented with ROCK inhibitor Y27632 (10 μM) for the first 24 h. The neuronal maturation medium was the same used for NPC-derived neuronal cultures. Cultures were maintained as previously described.

LSBX: LDN193189 (Stemgent, 250 nm), SB-431542 (Sigma-Aldrich, 10 μM) XAV939 (Sigma-Aldrich, 5 μM). PSD: PD0325901 (8 μM), SU5402 (10 μM), DAPT (10 μM). N-2 medium: DMEM/F12 with B-27 supplement (0.5×) and N-2 supplement (0.5×).

*NPC-derived astrocytes differentiation.* At differentiation day −1, 90–95% confluent NPC cultures were washed with PBS and incubated for 5 min at 37 °C with accutase. Accutase reaction was then stopped by resuspending NPCs in NPC medium followed by centrifugation at 300×*g* for 5 min. The supernatant was removed, and cells were resuspended by gently up and down pipetting to obtain a single-cell suspension. NPCs were then seeded in low-attachment 35-mm dishes and incubated shaking at 90 rpm for 24 h at 37 °C to obtain NPC spheres. The day after, NPC medium was replaced with DMEM-F12 supplemented with 1:100 B-27, 1:200 N-2, and 5 μM ROCK inhibitor. After 48 h, medium was changed to the astrocyte growth medium (AGM Bullet Kit, Lonza, #CC-3186) for 15 days, shaking at 90 rpm at 37 °C, changing medium every third day. After 2 weeks, the thus-obtained spheres were plated on poly-ornithine and laminin-coated dishes in astrocyte growth medium. When cells reach ~95% confluence, spheres were aspirated with a tip and adhered cells were passaged to a new dish. The culture was considered pure astrocytic after the third passage.

**Cerebral organoid culture.** Cerebral organoids were generated using a previously reported protocol[34]. Briefly, iPSCs at 70–80% confluence were detached by Accutase solution incubation at 37 °C for 10 min in order to obtain a single-cell suspension. Cells were centrifuged, counted, and a total of 9000 cells were then plated into each well of an ultra-low-attachment 96-well plate (Corning), in medium containing DMEM/F12, 20% KSR, 2 mM glutamine (Sigma-Aldrich),1% Pen/Strept, 1% nonessential amino acids, 50 nM β-mercaptoethanol (Thermo-Fisher Scientific) and 4 ng/ml bFGF. After seeding, plates were briefly centrifuged to allow single EB formation inside each well; ROCK inhibitor Y27632 (50 μM) was included in the first 24 h. EBs were maintained in 96-well plates for 6 days, then transferred by firmly pipetting (with a cut end of a P200 tip) medium in the well up and down to ultra-low-attachment 24-well plates (Corning), in neural induction medium containing DMEM/F12, 1× N-2 supplement, 1% nonessential amino acids, 2 mM glutamine, and 1 μg/ml heparin (Sigma-Aldrich). On day 10, EBs were embedded in droplets of Matrigel (Matrigel growth factor reduced, Corning), and were allowed to gel at 37 °C for 30–60 min. Embedded EBs were subsequently cultured in neural maturation medium containing 50% DMEM/F12, 50% Neurobasal A (ThermoFisher Scientific), 0.5× N-2 supplement, 0.5× B-27 supplement without vitamin A, 2 mM glutamine, 2.5 ng/ml human insulin (Sigma-Aldrich), 1% nonessential amino acids, and 25 nM β-mercaptoethanol. Droplets were cultured in the stationary condition in 6-cm suspension dishes for 4 days, followed by transfer to an orbital shaker (Orbit™ LS Low-Speed Orbital Shaker) rotating continuously at 60 rpm; here 0.5× B-27 supplement with vitamin A was substituted in the neural maturation medium. Samples for immunostaining were fixed in 4% paraformaldehyde for 20 min at 4 °C followed by washing in PBS three times 10 min. Tissues were allowed to sink in 30% sucrose overnight and then embedded in O.C.T. Compound (VWR) and cryo-sectioned at 20 μm.

**Organoid single-cell RNA-seq analysis**

*Sample preparation.* Three organoids per group (D868D and D868N) from the same batch at 4 weeks of culture were pooled and dissociated to single-cell suspension according to the protocol described in ref. [85]. Briefly, organoids were transferred to a Petri dish containing HBSS (+/+) using a 1-mL pipette tip and washed from eventual debris contained in culture medium. Organoids were then transferred to a 2-mL Eppendorf tube, where the remaining HBSS was removed. To dissociate organoids, Papain-based enzymatic digestion of tissue was applied using a Neural Tissue Dissociation Kit (P) (Miltenyi Biotec). Enzyme Mix 1 was prepared according to the manufacturer's instructions, pre-warmed for 15 min at 37 °C, then added to the organoids and incubated at 37 °C in a water bath for an initial 15 min. Enzyme Mix 2 (DNASe mix) was then added to the reaction. Organoid pieces were triturated (20×) first using a wide-bore pipette tip, and then a 1000p (1-mL pipette tip) to break the organoid into smaller tissue pieces. The tissue pieces were incubated for two additional 15 min intervals with intermediate (45 min in total) triturations using 1000p followed by 200p, until no tissue pieces or aggregates could be observed by eye. All the remaining clumps were filtered using a 30-μm cell strainer, to remove the remaining enzymes, 4 mL of HBSS were added to the pellet and suspended, spun down for 5 min at 300 × *g*. Percoll gradient centrifugation was applied to remove debris and enrich viable cells. This was performed by mixing HBSS with different volumes of a Percoll solution mix (Percoll, diluted in 10× HEPES solution (100 mM) to 1× HEPES): 125 μL Percoll dilution+1 mL HBSS (solution 1); 188 μL Percoll dilution+1 mL HBSS (solution 2), 250 μL Percoll+1 mL HBSS (solution 3), solution 3, 2, and 1 (1 mL each) were gently layered on top of each other in a 5-mL tube. The suspended cell suspension was gently pipetted on top of the gradient and spun down 5 min at 300×*g*. Debris-containing supernatant containing debris was discarded, and the healthy cell pellet was suspended in HBSS. Finally, the cells were counted using Trypan Blue and a Countess™ Cell Counter (ThermoFisher), and the viability was assessed to be greater than 85%. Dissociated cells were suspended in ice-cold HBSS with 0.04% BSA at a concentration of 1000 cells/μl. Single-cell libraries were prepared using Chromium Next GEM Single Cell 3' Kit v3.1 (10x Genomics) loading 5000 cells per lane.

*scRNA-seq analysis.* We processed raw paired-end scRNA-seq data using cellranger[86] (v4.0.0) (10× Genomics) with default parameters to generate the DGE matrices. We performed alignment to the GRCh38 reference genome using

cellranger[86] "count" with Gencode v37 annotations. We achieved unique mapping for around 90% of the reads in WT (D868D), and 90% of the reads in the mutant (D868N). We discarded non-uniquely mapped reads. To distinguish between those captured cellular transcriptomes from those that captured ambient RNA, we sorted barcodes by decreasing the number of reads and picked the inflection point ('knee') of the cumulative fraction of the reads plot. The cellranger outputs of each sample were then merged using cellranger function "aggr" with the corresponding replicate. We used Seurat[87] (v4.0) for downstream computational analyses. The two datasets were integrated first finding the so-called anchors shared between them using the FindIntegrationAnchors function and then merged with IntegrateData function. To remove damaged cells, we extracted the percentage of mitochondrial reads and the count of captured transcripts (nCount_RNA) and removed all barcodes with <200 nCount_RNA or high-percentage of mitochondrial reads (>20% in both WT and Mut). We removed barcodes with extremely low mitochondrial reads (<0.8%) to exclude nuclei. In order to exclude potential doublets, we also excluded cells with very high nUMI (>5000). With the previously described filtering, we were able to keep 4177 cells for WT condition and 1973 for Mutant one.

For each cell, UMI counts per gene were normalized and scaled. We performed clustering considering only the top 2000 highly variable genes, as identified by the function "FindVariableGenes". Variable genes were then used to perform principal component (PC) analysis. We selected the PCs to be used for downstream analyses by evaluating the "PCEl-bowPlot" and the "JackStrawPlot". We used the first 15 PCs for both Wt and Mut. We identified clusters using the function "FindClusters", which exploits a SNN modularity optimization clustering algorithm (at Resolution = 0.7). We visualized clusters using the uniform manifold approximation and projection (UMAP) dimensionality reduction[36]. We used the manual inspection of marker genes determined using the "FindAllMarkers" function for cluster identification. This function determines which genes, that are expressed in at least three cells, are enriched in every clustering using log2FC threshold values of 0.25 and 0.05 of adjusted *P* value (FDR). After genes extraction for all clusters, we assigned each cell to a cell cycle phase (G1, 2, G2M) using the "CellCycleScore" function and a published gene set[88], we normalized the numbers of cells for each specific phase by dividing with the total amount of cell within the respective condition.

Raw data have been deposited at NCBI GEO: GSE171266 superseries, scRNAseq in GSE171263 subseries.

**Cortical spheroid culture.** Cortical spheroids were generated using a previously reported protocol[35]. Briefly, iPSCs at 70–80% confluence were detached by Accutase solution incubation at 37 °C for 10 min in order to obtain a single-cell suspension. Cells were centrifuged, counted, and a total of 30,000 cells were then plated into each well of an ultra-low-attachment 96-well plate (Corning), in mTeSR1 supplemented with ROCK inhibitor Y27632 (50 μM); after seeding, plates were briefly centrifuged to allow single EB formation inside each well. After 24 h, spheroids from each well were collected by firmly pipetting (with a cut end of a P200 tip) medium in the well up and down and transferring it into 6-cm low-adhesion plastic dishes in medium containing DMEM/F12, 20% KSR, 2 mM glutamine (Sigma-Aldrich),1% Pen/Strept, 1% nonessential amino acids, 50 nM β-mercaptoethanol (ThermoFisher Scientific) supplemented with two SMAD pathway inhibitors—dorsomorphin (2.5 μM, Sigma-Aldrich) and SB-431542 (10 μM, Sigma-Aldrich). From day 2 to day 6, medium was changed every day and supplemented with dorsomorphin and SB-431542. On the 6th day in suspension, neural spheroids were transferred to neural medium containing Neurobasal A (ThermoFisher Scientific), B-27 supplement without vitamin A (ThermoFisher Scientific), 2 mM glutamine (Sigma-Aldrich) and 1% Pen/Strept. The neural medium was supplemented with 20 ng/ml epidermal growth factor (EGF, ThermoFisher Scientific) and 20 ng/ml bFGF for 19 d (until day 24), with medium changed daily in the first 10 d and every other day for the subsequent 9 d. To promote differentiation of the neural progenitors into neurons, the neural medium was supplemented with 20 ng/ml BDNF and 20 ng/ml NT3 (Peprotech), with medium changes every other day. Samples for immunostaining were prepared as previously described for cerebral organoids.

**Cell treatments.** To downregulate AKT signaling in NPCs MK-2206 dihydrochloride (5 μM, AKT inhibitor, MedChemExpress) was added to the differentiation medium of NPCs from iPSCs, starting from Rosette stage, and kept for 12 days of culture, freshly added with every medium change. Vehicle (0.05% DMSO, Sigma-Aldrich) was added to both D868D and D868N cells as an experimental control. Cells for subsequent analysis were collected at "NPC p5" (Fig. 4h–j). To overstimulated TP53 signaling in NPCs, 1 μM doxorubicin (Sigma-Aldrich) was added to NPC culture medium of both D868D and D868N cells for the indicated time points (Fig. 5e and Supplementary Fig. 6e). For increasing TP53 protein, Nutlin-3a (Sigma-Aldrich) was added to differentiation medium of NPCs from iPSCs, starting from the Rosette stage, and kept for 12 days of culture, freshly added with every medium change. Vehicle (0.025% DMSO, Sigma-Aldrich) was added to both D868D and D868N cells as an experimental control. Cells for subsequent analysis were collected at "NPC p5" (Fig. 5f, g and Supplementary Figs. 7 and 12d). For the inhibition of PARP−1, 1 μM Olaparib (Selleckchem) was added to the different differentiation media during the whole differentiation process from iPSCs to neurons, at every medium change for D868N cells. Vehicle

(0.01% DMSO, Sigma-Aldrich) was added to both D868D and D868N cells as experimental control (Fig. 8a). Also, other inhibitors of cell death pathways (Fig. 8c) were delivered at the same conditions: Z-VAD(OMe)-FMK(Z-Val-Ala-Asp(OMe)-FMK) for apoptosis inhibition (MedChemExpress, 10 μM) and Necrostatin 2 racemate (Nec-1S) for necroptosis inhibition (MedChemExpress, 10 μM). Staurosporine (Sigma-Aldrich) was added to the neuronal culture medium as a positive control for apoptosis activation for 12 h at 2 μM (Supplementary Fig. 10f). β-Nicotinamide adenine dinucleotide hydrate (Sigma-Aldrich, 500 μM) was delivered to D868N NPC-derived neuronal cultures in order to rescue NAD deficiency and neuronal mortality starting from day 4 of neuronal differentiation and at every medium change (Fig. 8e).

**NPC proliferation rate**. To measure the NPCs proliferation rate, $3 \times 10^5$ cells were seeded on Matrigel-coated 12-multiwell plates. Twice a week, cells were detached, live cells were stained with Trypan Blue Solution (0.4%, ThermoFisher Scientific) and counted using Countess™ II Automated Cell Counter (ThermoFisher Scientific); after this passage, $3 \times 10^5$ cells were seeded again. This was repeated for six time points; the experiment was repeated three times for each time point. Population doublings were calculated using the equation: population doubling (PD) = log10($N$)/log10(2), where $N$ is the ratio between the number of cells harvested at the end of the culture and the number of seeded cells. Cumulative PD (CPD) was calculated for each passage as the sum of the current and all the previous PD values.

**Flow cytometry**. For cell cycle analysis, NPCs were seed on matrigel-coated plates at a density of 80,000 cells/cm$^2$. Twenty-four hours after seeding, the cells were detached with Accutase solution, harvested, and fixed in 70% ethanol on ice for 2 h; after washing with PBS, the fixed cell was treated with RNase A (200 μg/ml, Carlo Erba) and stained with propidium iodide (50 μg/ml, ThermoFisher Scientific) at 37 °C in the dark for 20 min. Samples were immediately acquired on FACS Canto II (BD) flow cytometer, and cell cycle profiles were analyzed using FCS Express 6 Flow (De NovoSoftware) and expressed as a percentage. In order to analyze G1/S exit, cells were synchronized by double thymidine block and analyzed by FACS at different time points after thymidine release[89]. Briefly, 24 h after seeding thymidine (2 mM, Sigma-Aldrich) was added to the culture medium, and cells were cultured for 18 h; then thymidine was removed and cells were released for 9 h in fresh NPC medium. The second round of thymidine (2 mM) was added for 18 h. Then cells were released by washing with 1× PBS and incubating them in fresh NPC medium. Cells were collected at 0, 2, 10, and 24 h after release for propidium iodide staining and FACS analysis as described before.

**RNA isolation and real-time RT-PCR**. RNA was extracted using the TRI Reagent isolation system (Sigma-Aldrich) according to the manufacturer's instructions. For quantitative RT-PCR (qRT-PCR), one microgram of RNA was reverse transcribed using the ImProm-II Reverse Transcription System (Promega), thereafter qRT-PCR was performed in triplicate with custom-designed oligos (see Supplementary Dataset 4) using the CFX96 Real-Time PCR Detection System (Bio-Rad, USA). using the Titan HotTaq EvaGreen qPCR Mix (BIOATLAS). Obtained cDNA was diluted 1:10, and was amplified in a 16 μl reaction mixture containing 2 μl of diluted cDNA, 1× Titan HotTaq EvaGreen qPCR Mix (Bioatlas, Estonia), and 0.4 mM of each primer. Analysis of relative expression was performed using the ΔΔCt method, using 18S rRNA as housekeeping gene and CFX Manager software (Bio-Rad, USA).

**RNA-Seq**. Total RNA was extracted as previously described. Sequencing was performed by GENEWIZ sequencing company (Germany). FASTQ reads were quality checked and adaptor trimmed with FastQC (Andrews, S. FastQC A Quality Control tool for High Throughput Sequence Data). High-quality trimmed reads were mapped to the hg38 reference genome with the STAR aligner[90] (v2.7.9.a) using the latest GENCODE main annotation file[91]. Differential gene expression was calculated with DESeq2[92] (v1.32). Geneset functional enrichment was performed with GSEA[93] (v4.1.0). Downstream statistics and Plotting were performed within the R environment. Heatmaps were generated with GENE-E (v3.0.215) (GENE-E. Cambridge (MA): The Broad Institute of MIT and Harvard). RNA-Seq data from the literature were downloaded from the NCBI GEO repository with the accession codes provided in Supplementary Dataset 5. Our raw data were deposited at NCBI GEO: GSE171266 superseries, GSE150810 bulk RNA-seq series.

**Immunostaining**. Cells were seeded on coated glass coverslips and they were fixed for 20 min on ice in 4% paraformaldehyde (PFA, Sigma), solution in phosphate-buffered saline (PBS, Euroclone). Then they were washed twice with PBS and were permeabilized for 30' in blocking solution, containing 0.2% Triton X-100 (Sigma-Aldrich) and 5% donkey serum (Euroclone), and incubated overnight at 4 °C with the primary antibodies diluted in blocking solution according to information in the antibody datasheet (see Supplementary Dataset 5 for the complete list of primary antibodies used and their working dilution). The next day, cells were washed three times with PBS for 5 min and incubated for 1 h at room temperature with Hoechst 33342 (ThermoFischer Scientific) and with secondary antibodies (ThermoFisher Scientific) in blocking solution. Organoids slice frozen sections underwent the same

procedure for immunostaining. Images were acquired with epifluorescence microscope Nikon DS-Qi2 and analyzed with Fiji software. For 2D culture immunostaining, replicates showed in the bar graphs represent the number of coverslips from independent platings considered for each quantification.

*Quantification of immunostaining in 3D cultures*. Organoid shape at 4 weeks was quantified by measuring the circularity of the external perimeter in serial organoid slices. Circularity is a normalized ratio of area to the perimeter (circularity = 4pi (area/perimeter$^2$) with a circle having a circularity of 1 and lines having a circularity of 0[94]. Other parameters (length of rosette lumen, number of rosettes per organoid, neuroepithelial thickness, presence of DNA damage and cell death, proliferation) were measured in representative slices of different organoids from different batches. The number of ventricles (v), organoids (o), and batches (b) considered for each quantification are specified in figure legends.

**Nuclear shape**. The nuclear shape was analyzed by measuring nuclei circularity with Fiji software[94] (v1.48).

**Sholl analysis**. Neuronal cultures derived from NPCs were transduced with lentiviral vector EF1a-GFP at a low titer at day 4 of differentiation for 1 h in neuronal maturation medium, in order to obtain sparse GFP cell-labeling. At 4 weeks of differentiation, cells were processed for IF analysis as previously described; images of the dendritic tree of double-positive GFP + /MAP2 + cells were analyzed using Sholl Analysis plugin[95] in Fiji software (v1.48) (NIH, USA).

**Neuronal death assay**. Neuronal cultures at 4 weeks were live-stained using Propidium Iodide (ThermoFisher Scientific, 1 μg/μl) and RNase A (100 μg/ml, Sigma-Aldrich) solution in PBS at 37 °C in the dark for 20 min. Cells were subsequently processed for IF analysis, as previously described.

**TUNEL assay**. TUNEL assay was performed using DeadEnd™ Fluorometric TUNEL System (Promega) on 8 week-old organoid slices following the manufacturer's instructions.

**Single-cell gel electrophoresis (SCGE or comet assay)**. Comet assay in NPCs was performed using CometAssay HT kit (Trevigen) in alkaline conditions, following the manufacturer's instructions. WT NPCs treated with 10 μM doxorubicin (Sigma-Aldrich) for 24 h were used as a positive control for the assay. Images were analyzed with Open Comet plugin[96] in Fiji software (v1.48i) (NIH, USA). At least 70 cells per group from three different experiments were analyzed.

**Western blot analysis**. iPSCs, NPCs, neurons, cerebral organoids or spheroids (pool of 10/15 organoids from the same batch) were homogenized in RIPA buffer (50 mM Tris pH 7.5, 150 mM NaCl, 1 mM EDTA, SDS (0.1% for cells, 1% for 3D cultures), 1% Triton X-100, Roche Complete EDTA-free Protease Inhibitor Cocktail, Roche PhosSTOP EASYpack) and western blot analysis was performed incubating primary antibodies overnight at 4 °C in blocking solution composed by 5% BSA (Sigma-Aldrich) or 5% nonfat dry milk in PBS-TWEEN 0.1% (Sigma-Aldrich) according to antibody datasheet (see Supplementary Dataset 5 for the complete list of primary antibodies used and their working dilution). Band densitometry relative to control was calculated using Fiji software (v1.48i) (NIH, USA), normalized on housekeeping as indicated in each figure (H3, CALNEXIN, β-ACTIN); post-translational modifications were analyzed by first normalizing both total and a modified form of the protein of interest on housekeeping, and then plotting the ratio between modified vs. total relative to control. Replicates showed in the bar graphs represent the number of blot from different cell lysates from independent experiments considered for each quantification. For 3D cultures, lysates originate from at least three different batches.

**PP2A activity assay**. PP2A phosphatase activity was measured in iPSC, NPC, or neuronal cultures using PP2A Immunoprecipitation Phosphatase Assay Kit (Sigma-Aldrich) according to the manufacturer's instructions.

**NAD/NADH detection assay**. The ratio between total oxidized and reduced nicotinamide adenine dinucleotides (NAD + and NADH, respectively) was detected and determined in NPC-derived neuronal cultures using the NAD/NADH-Glo™ Assay (Promega) according to the manufacturer's instructions.

**Quantification and statistical analysis**. Data in figure panels reflect several independent experiments performed on different days. No data were excluded. No statistical methods were used to predetermine sample size in other experiments. Samples were not subject to randomization but were assigned to experimental groups based on their genotype. Data are expressed as mean ± standard error (SEM) and significance was set at $P < 0.05$ (see each figure for details). Statistical details are provided along the "Methods" section (procedure) and either in the figure legends or in Supplementary Dataset 3 (values). Differences between means were analyzed using the Student's two-sided unpaired $t$ test. $t$ test and one-way or

two-way ANOVA depending on the number of groups and variables in each experiment. Data were then submitted to Tukey post hoc test using GraphPad Prism software. The null hypothesis was rejected when $P < 0.05$. In the graphs, "ns" indicates nonsignificant differences when $P > 0.05$, * indicates significant differences with $P < 0.05$, ** indicates significant differences with $P < 0.01$, *** indicates significant differences with $P < 0.001$ and **** indicates significant differences with $P < 0.0001$. Micrographs in Figs. 2a, 6c, 7a, and Supplementary Figs. 1c, 8a-b, 8h, 10f, and 12f are representative of iPSCs from passages 25–50 (Supplementary Fig. 1c) from three different clones per line and three coverslips from independent platings for each sample, NPCs from passages 3–10 from five independent differentiation experiments (Fig. 2a), neurons from three independent differentiation experiments (Fig. 7a and Supplementary Fig. 10f, 12f), and three independent batches of cerebral organoids (Fig. 6c and Supplementary Fig. 8a, b, h).

**Reporting summary**. Further information on research design is available in the Nature Research Reporting Summary linked to this article.

## Data availability

All datasets generated in this work are available at the NCBI Gene Expression Omnibus (GEO) database: "GSE171266". All other relevant data supporting the key findings of this study are available within the article and its Supplementary Information files or from the corresponding author upon reasonable request. Source data are provided with this paper.

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

## Acknowledgements

We thank Prof. Alexander Hoischen for sharing the fibroblast from patients, them, and their families. We thank the members of the Broccoli lab for helpful discussions, as well as Dr. Dario Bonanomi for critical reading of the manuscript. A.S. was supported by Telethon (GGP15096), the Italian Ministry of Health (GR-2013-02355540; GR-2019-12370949), and Fondazione Regionale per la Ricerca Biomedica (Regione Lombardia), European Joint Programme on Rare Disease Project TREAT-SGS (EJPRD20-008), *GA* 825575. V.B. was supported by Italian Ministry of University and Research (PRIN2017 # 2017M95WBA). R.P. was supported by AIRC (AIRC IG-22082). We thank SGS foundation for continuous support and help. Drawings were made with BioRender.com.

## Author contributions

A.S. conceived this study. F.B., V.B., and A.S. designed the experiments. F.B. performed and analyzed all the experiments with the help of A.R., M.Z., G.F., M.L., C.C., C.D.R., C.M., and A.I.; A.B., M.F., R.P., and L. Mologni provided materials and advice. L. Massimino and E.B. performed bioinformatics analysis. A.S. wrote the manuscript with inputs from F.B., A.R., M.Z., L. Massimino, C.D.R., R.P., L. Mologni, and V.B.

## Competing interests

The authors declare no competing interests.
