## [Peer Review File · Nature Communications]

Reviewers' Comments:

Reviewer #1:

Remarks to the Author:

SGS (Schinzel-Giedion syndrome) is rare genetic disorder, with neurodegeneration at early age of patients. The authors derived iPSC lines from SGS patients that have mutations (I871T, D868N) in SETBP1, and performed CRISPR-gene editing to isolate so-called isogenetic control iPSC lines. Authors applied 2D neural differentiation, 3D brain and cortical organoid differentiation, and NGN2-induced neural differentiation to study the function of SETBP1 mutants in brain development and degeneration. They found that SETBP1 mutations cause the increase in cell proliferation, and DNA damage in NPCs but without cell death. By directly generating neurons by NGN2 induction, authors found that the post-mitotic neurons from mutant iPSCs undergo cell death via PARP1-dependent accumulation of PAR and thus parthanatos. While the concepts are interesting and novel, a number of conclusion from biochemical experiments are fully supported by the results, and quantification of data are not of high standard.

1. In major of quantification of Western blot does not seem to represent what is shown in Western blot figure. In figure 1D, the high molecular weight bands of SET for D868D and D868N are less than WT. In figure 1E and F, the pPP2A/PP2A seem lower in D868N and I871I compared with WT. In figure 2E, the phosphorylated PP2A forms seem higher in mutants compared with WT. Overall, authors should show data for all repeats of the Western blot to show whether the quantifications in graph are supported by the data. One of the major conclusions that PP2A activity does not change in the mutations in SETBP1 and thus different from phenotypes in the myeloid cells is not supported by the data.

2. Along the same line, in figure 1G, the Immunofluorescence results seem to show that the number of cells showing the pH3 of D868N is higher in D868N compared to control. However, authors conclude that there was no difference. Authors need to perform more experiments to perform statistical analysis to make the conclusion.

3. In figure 4B, again the quantification of the Western blot does not faithfully represent the blot data. I871I and D868D seem to express lower pH2AX than the I871T and D868N and WT. It looks like that when authors derive the isogenic iPSC lines by performing CRISPR editing, the derived clones are less sensitive to DNA damage. During CRISPR editing, DNA is cut and repaired and thus the clones that were selected had better DNA repair capability. Thus, the DNA damage assay here may not be best methods to address whether I871T and D868N mutants are DNA damage sensitive. Authors may perform the experiments using plain WT as control to see whether the plain WT show similar trends in DNA damage response like I871T and D868N mutants, or CRISPR-edited I871I and D868D mutants. This also apply to figure 4E and 4F for p53 response.

4. In introduction, there is not enough information about SETBP1. Authors need to describe additional SETBP1 information such as molecular function or biological process.

5. In figure 2D, the authors distinguished between the low molecular weight band and the high molecular weight band in figure 1D, but not here. Make sure to show this result as shown in figure 1D. In addition, in figure 2H and S2B, there are no figure data on I871T and I871I as well as plain WT. Authors need show the data for theses to support the conclusion.

6. In figure 3 and 5, the authors have identified AKT protein expression in NPCs or cerebral organoids. Authors should examine the expression of the AKT signal-related protein such as mTOR, pS6 or NF- κ B protein? Certainly in order to show that the signal is regulated, the authors have to show that the related protein is regulated experimentally.

Reviewer #2:

Remarks to the Author:

The manuscript by Banfi et cols presents a study of the molecular mechanisms underlying neurodegeneration in Schinzel-Giedion syndrome (SGS). The authors have generated patient-

derived induced pluripotent stem cells (iPSCs) to model neurogenesis in this disorder and performed extensive characterization of the molecular alterations arising during this process. The main conclusions of the study are the following:

- The authors establish a cellular model for SGS using patient-derived iPSCs and isogenic controls, from which neural progenitor cells, cortical neurons and brain organoids are derived.
- The presence of mutations in the SETBP1 degron during neurogenesis leads to an increase in SETBP1 and SET protein in NPCs, inhibiting P53 acetylation and activation.
- NPCs derived from SGS patient iPSCs show accumulation of DNA damage, which is inherited by post-mitotic neurons. This causes increased neuronal death through a PARP1-mediated mechanism.

The study is a detailed analysis of different alterations observed in a neurogenesis model for SGS, providing mechanistic insights into the process of neurodegeneration in this disease (and potentially into options for therapeutic development). It was a pleasure to review this manuscript, as it presents novel and very interesting findings with great relevance for both neurodevelopmental disease and cancer. However, additional data is required to better establish the link between the molecular alterations observed and fully support the molecular mechanisms described in the manuscript.

Major comments

1. Figures 2G-H show that SGS NPCs proliferate more than isogenic controls. Additionally, in Figure 2I, the authors show an increase in the proportion of SGS NPCs in S and M/G2 phase and a decrease in the proportion of SGS NPCs in G1 phase compared to isogenic controls. Given that SGS NPCs show: 1) decreased expression for genes involved in mitotic cell cycle arrest and upregulation of genes involved in proliferation (RNA-seq data in Figure 3); and 2) alterations in P53 (Figure 4E), which controls cell cycle arrest, this finding is worth exploring in more detail. For example, the authors could provide additional data to document alterations in progression through cell cycle checkpoints such showing whether SGS NPCs exit G1 phase of the cell cycle faster than isogenic controls (particularly in the presence of unresolved DNA damage).

2. The authors show evidence for dysregulation in the expression of genes involved in the cell cycle (Figure 3), but it is not clear whether this is caused directly by SETBP1 (as a consequence of SETBP1's role in activating gene expression, as described in Piazza et al. 2018), as a consequence of P53 alterations due to inhibition of acetylation or to further downstream effects. Could the authors provide further detail on this?

3. PTEN is shown to be downregulated in Figure S3E and suggested to be connected to an increase in pAKT and hyperactivation of this pathway. The alteration in the AKT/PI3K pathway is significant, as an increase in AKT/PI3K signaling could lead or contribute to inhibition of cell death and increased cell proliferation (as shown in Figure 2G and 2H). As this pathway is involved in various cellular processes in neurogenesis and has been implicated in other neurodevelopmental disorders, it would be valuable to establish this link with further data (e.g. showing a decrease in PTEN at the protein level or an increase in PIP3 levels which would support a link between both observations, examining the effect of AKT signaling on cell proliferation). Given the role of AKT/PI3K pathway in neuronal migration, this finding is also interesting in light of previous results which have shown defects in neuronal migration in mouse models with SETBP1 mutation (Piazza et cols. 2017). Could the authors please comment on this?

4. Previous research has shown that SET protein binds P53, hampering its acetylation and activation. SGS NPCs have increased SET protein levels, a decrease in P53 acetylation (Figure S4D) and dampened P53 acetylation in response to DNA damage induced by doxorubicin (Figure 4E). From the data presented, it is not clear whether the disruption to P53 acetylation observed in SGS NPCs causes the other molecular and cellular alterations observed. To examine this, the authors could test whether increasing levels of P53 protein in SGS NPCs restores levels of acetylated P53 protein and consequently other alterations observed in these cells (e.g. cell cycle alterations, correcting the dysregulation of some genes, increasing apoptosis, decreasing PARP-1 activation, etc).

5. The authors generate brain organoids from the SGS IPCs (Figure 5), which reproduce molecular alterations observed in the 2D cell cultures. Furthermore, the brain organoids present an increase in folds comparable to that observed in organoids derived from cells with PTEN loss-of-function. In the paper referenced by the authors (Li et al 2017), the PTEN mutant organoids are treated with inhibitors of AKT activation to determine whether AKT activation is responsible for the expansion and folding phenotypes. Based on these results, it would be interesting to understand the role of AKT activation in the abnormalities observed in the SGS brain organoids by treating these organoids with inhibitors as described in the above-mentioned paper.

6. The authors show that inhibition of PARP-1 reduces DNA damage in SGS NPCs (Figure S7A), which suggests that PARP-1 activation may be contributing to DNA damage potentially as a result of NAD⁺ depletion. This is supported by data in Figure 7D showing decreased NAD⁺/NADH in SGS neurons. Can the authors comment on whether PARP-1 hyperactivation and NAD⁺ depletion could contribute to the severe epilepsy observed in patients with SGS? Furthermore, could the authors comment on the expected effect of PARP-1 inhibition in post-mitotic neurons (i.e. after accumulation of DNA damage during neurogenesis) as a potential therapeutic approach?

7. It is difficult to follow the relationship between the molecular and cellular abnormalities described by the authors in this manuscript. It would be beneficial to include a drawing or graphical representation to summarize the molecular alterations observed and how they relate to each other.

Minor comments:

1. Please revise the manuscript for language and errors in grammar. Some sentences convey seemingly incorrect information (e.g. p10, line 298: "Of note, the usage of either the broad-spectrum caspase inhibitor carbobenzoxy-valyl-alanyl-aspartyl-[O-methyl]-fluoromethylketone (Z-VAD) or the necroptosis inhibitor Nec-1s had appreciable effects (Figure 7C)." whereas Figure 7C suggests otherwise). Some sentences are difficult to understand (e.g. p18, line 524: "(C) Histograms showing the number of GO categories relative to the indicated different cell types retrieved in genes upregulated in controls (dark green) and mutant NPCs (light green)."). Additionally, there are minor errors (e.g. p2, line 29: "... represent informative cases for shed light on the causative mechanisms.") and typos (e.g. p29, line 903: "...analyzed by immunofluoresce").

2. In the manuscript abstract (p2, line 30), it would be preferable to state that SGS is caused by SETBP1 mutations (which then lead to the accumulation of SETBP1 protein), especially since tissue-specific SETBP1 mutations are mentioned further on in the abstract.

3. In addition to establishing iPSCs from patients with SGS, the authors derive isogenic control cells by mutating the SGS allele into a wild-type allele. According to Figure 1B and S1A, this was done by targeting the locus with CRISPR and inducing homology directed repair although no information is provided in the Methods on the homologous repair template. Can the authors provide more information on the homologous repair template used (e.g. ssODN, additional plasmid, endogenous copy of the wild-type allele...)? Additionally, the methods state that double antibiotic selection was applied until complete death of non-transfected cells. The assumption is that the antibiotic selection was short to select for transfected cells (rather than for cells with genomic integration of the plasmids), but the Methods would benefit from clarification by specifying the duration of the antibiotic selection.

4. Figure 2C shows that NPCs derived from SGS iPSCs have significantly higher SETBP1 protein levels than NPCs from their isogenic controls. However, the graph shows a relatively large bar for the SEM in the quantification of the SETBP1 protein by Western blot visible across the WT, D868N and I871T samples. It would be interesting to understand whether this observation is due to technical/ experimental error (e.g. limitation in the precision of protein quantification with Western blot) or if there are significant differences in the levels of SETBP1 protein in the same cell line across different experiments (e.g. SGS D868N NPCs in experiment #1 versus experiment #2). In relation to this, a question that comes to mind is whether differences in the levels of SETBP1 protein between samples of the same cell line may reflect differences in the level of differentiation of the cells (between iPSCs and NPCs) or differences in the proportion of cells in a specific phase

of the cell cycle. Could the authors comment on this?

5. In Figure 2I and 3G, results are shown for D868N #1 and #2 which likely represent technical replicates, but this is not stated in the legend or text.

6. Figure S1A is too small to read the text. Please make larger or increase the font size. Additionally, Figure S1C shows data for isogenic wild-type cells with mutation I871N. Please correct or clarify.

7. Figure S3A shows over- and under-expressed genes in D868N and I871T cells in a Venn diagram. This type of diagram is not optimal to display the data, as there is no overlap expected between over-expressed and under-expressed genes in the same experimental sample.

Reviewer #3:

Remarks to the Author:

In this manuscript, Banfi and colleagues study the role of 2 mutant variants of SETBP1, naturally causing Schnitzel-Giedin Syndrom in humans. This is a rare neurodevelopmental disorder that leads to premature neurodegeneration at the early stages. The authors promise in the abstract that studying this specific disease will shed new light on the onset of neurodegenerative diseases, which is not the case. Nevertheless, they characterize the NPCs and neurons derived from these patients, and convincingly show an interesting mechanism underlying this pathology.

I found this manuscript well-written and quite easy to follow. The experiments are justified even though sometimes the logical links between steps are weak.

Specific Comments:

The authors are quite vague on replicates and numbers, they should be more precise about the number of coverslips, technical and biological (batches) replicates, the number of organoids, ventricles, cells analyzed for each experiment since the variability of these systems is very high.

I find it quite weird that the genes upregulated in mutant NPCs are associated with different lineages. Did they reproduce the data with different differentiation batches?

Can the authors comment a bit more clearly to which type of cells the mutant NPCs are similar to?

The organoid data are very vague, how did the authors quantify the "more elaborated" mutant organoids? What does this mean? How did they quantify the folds?

How was the "increase of neuroepithelial like structure" quantified? The number of ventricles? Per organoid? The authors should show the entire organoid with some tissue-clarity method to show this change.

How was the neuroepithelial layer expansion assessed? Ventricular length? Thickness? Number of DAPI+ cells at the ventricle?

Why cortical spheroids did not fold? Are other (non-dorsal) cells necessary for the formation of the folds?

Where did they show that hyperproliferation occurs in spheroids?

The authors then wanted to assess the mutant function in neurons. Why did they artificially express Ngn2? This is not the default protocol to generate neurons, why did they want to speed up the generation of neurons? Can the authors reproduce these data by only changing the medium without forcing the expression of Ngn2 to generate different types of neurons?

Can the authors reproduce the neuronal phenotype in the more physiological 3D organoids?

The authors have invested already in generating cerebral organoids, can they assess the neuronal phenotype in cerebral organoids as well by looking at later stages?

The function of this protein seems to be quite cell-specific, could the author investigate different

neuronal excitatory subtypes, interneurons, glial cells to connect the phenotype to something more concrete with patients?

The authors provide evidence that this is the mechanism underlying this neurodegenerative disease with the idea that this could shed new light on others. what is their hypothesis now?

I wonder if this hyperproliferation followed by neuronal death is something that can be seen in patients. In light of the massive phenotype regarding proliferation at initial stages, it may be possible to correlate this with an increase in brain size at early stages? Is the correlation with a decrease at later stages?

Single-cell RNA-seq in organoids would be the best way to figure out all these details unbiasedly.

Point by Point Response about our MS NCOMMS-20-34839-T by Banfi et al.

Reviewer #1 (Remarks to the Author):

SGS (Schinzel-Giedion syndrome) is rare genetic disorder, with neurodegeneration at early age of patients. The authors derived iPSC lines from SGS patients that have mutations (I871T, D868N) in SETBP1, and performed CRISPR-gene editing to isolate so-called isogenic control iPSC lines. Authors applied 2D neural differentiation, 3D brain and cortical organoid differentiation, and NGN2-induced neural differentiation to study the function of SETBP1 mutants in brain development and degeneration. They found that SETBP1 mutations cause the increase in cell proliferation, and DNA damage in NPCs but without cell death. By directly generating neurons by NGN2 induction, authors found that the post-mitotic neurons from mutant iPSCs undergo cell death via PARP1-dependent accumulation of PAR and thus parthanatos. While the concepts are interesting and novel, a number of conclusion from biochemical experiments are fully supported by the results, and quantification of data are not of high standard.

We thank the reviewer for the general appreciation of our work and for the suggestions that have improved the quality of the revised work. Indeed, following the reviewer's recommendations, we added this set of new results:

- the biochemical experiments, in which we have repeated some blot, included unrelated WT, ameliorated quantifications and provided both full blots and the blots used for the presented quantifications;
- detailed analysis of the AKT signaling, providing new evaluations of protein levels and using inhibitors to manipulate the signaling.

Moreover, we provided a deeper investigation of the cellular phenotype downstream to P53 enhancement and a new thorough characterization of brain organoids, including new measurements, scRNAseq and generation of more mature organoids.

here below you can find a point-by-point response to the issues addressed and please note that we highlighted the changes in the manuscript using red sentences.

We hope that the new version of the manuscript has eliminated the weaknesses highlighted by this reviewer.

1. In major of quantification of Western blot does not seem to represent what is shown in Western blot figure. In figure 1D, the high molecular weight bands of SET for D868D and D868N are less than WT. In figure 1E and F, the pPP2A/PP2A seem lower in D868N and I871 compared with WT. In figure 2E, the phosphorylated PP2A forms seem higher in mutants compared with WT. Overall, authors should show data for all repeats of the Western blot to show whether the quantifications in graph are supported by the data. One of the major conclusions that PP2A activity does not change in the mutations in SETBP1 and thus different from phenotypes in the myeloid cells is not supported by the data. Unfortunately, the protein levels across different isogenic pairs may result quite different so sometime the representative blot we showed could be confusing. In the new version of the manuscript we have substituted blots in figures 1d, 1e, 2e. As suggested by the reviewer we provide all the blots used for the indicated quantification (Fig. for the reviewers 1-3) as well as the full blots of the crop images inserted in the figures (Fig. S13-16). For the matter of clarity, in the new version of the manuscript we have also enclosed different quantifications merging all control samples (meaning unrelated WT and isogenic controls) and all mutants (the two different SETBP1 mutations), specifically in the figures 1c, S1f (related to 1d), 1e, S1g (related to 1g), 2c, 2d, S2e (relative to 2e), s2f (relative to 2h), 4b, and S11c. Regarding PP2A in iPSC (Figure 1e and f), we admit that the ratio is variable between sample and replicates, however we never saw a clear evidence toward increased phosphorylated levels in mutants as expected if PP2A is inactivated (see Fig 5b in Piazza et al., Nat Gen, 2013). We have substituted the blot in the figure 1e with one of better quality. However, in our work we didn't rely only on pPP2A levels in WB but also on an independent measurements of PP2A activity using the PP2A IP Phosphatase Assay kit (Millipore) (see Fig 5c in Piazza et al., Nat Gen, 2013) that never indicated reduced activity in neither iPSC (Fig. 1f) nor NPCs (Fig. 2f).

2. Along the same line, in figure 1G, the Immunofluorescence results seem to show that the number of cells showing the pH3 of D868N is higher in D868N compared to control. However, authors conclude that there was no difference. Authors need to perform more experiments to perform statistical analysis to make the conclusion.

For this experiment, we performed 4 independent replicates for each condition. In the revised manuscript we added new images for the I871I and I871T lines. For a more comprehensive understanding we've now presented low magnification images that visualize the entire colonies.

3. In figure 4B, again the quantification of the Western blot does not faithfully represent the blot data. I871I and D868D seem to express lower pH2AX than the I871T and D868N and WT. It looks like that when authors derive the isogenic iPSC lines by performing CRISPR editing, the derived clones are less sensitive to DNA damage. During CRISPR editing, DNA is cut and repaired and thus the clones that were selected had better DNA repair capability. Thus, the DNA damage assay here may not be best methods to address whether I871T and D868N mutants are DNA damage sensitive. Authors may perform the experiments using plain WT as control to see whether the plain WT show similar trends in DNA damage response like I871T and D868N mutants, or CRIPSR-edited I871I and D868D mutants. This also apply to figure 4E and 4F for p53 response.

We thank the reviewer for pointing out this issue. It is important to underline that both control and mutant isogenic lines underwent CRISPR treatment. In fact, we selected unedited (or faithfully repaired) cells as D868N and I871T mutants while successfully edited clones were selected as control lines (D868D and I871I) in our experiment. Having said so, it

remains the possibility that mutants, if unedited, may display less DNA repair capability of the isogenic controls. For this reason, we have now included unrelated WT cells as controls in comet assay experiment and in p53 rescue experiment (Figure 4), as requested. However, from our data isogenic control and unrelated WT have the same capability of DNA repair. Moreover, we also showed that at iPSC stage the level of DNA is not statistically different among the lines (Fig. S11c).

4. In introduction, there is not enough information about SETBP1. Authors need to describe additional SETBP1 information such as molecular function or biological process.

We thank the reviewer for this suggestion. We added a more comprehensive presentation of the different SETBP1 functions in physiological and pathological conditions. In particular, we elaborated on the SETBP1 function in controlling the expression of different oncogenic factors, especially in the context of hematopoiesis and on the recent demonstration that can act as part of an epigenetic activator complex.

5. In figure 2D, the authors distinguished between the low molecular weight band and the high molecular weight band in figure 1D, but not here. Make sure to show this result as shown in figure 1D. In addition, in figure 2H and S2B, there are no figure data on I871T and I871I as well as plain WT. Authors need show the data for these to support the conclusion. The reviewer is right in this consideration. We've added the quantification of the upper band (that however is very faint in this cell type). As suggested we added the figures regarding the isogenic pairs of the mutation in I871 and the unrelated WT in Fig 2h and Fig S2b.

6. In figure 3 and 5, the authors have identified AKT protein expression in NPCs or cerebral organoids. Authors should examine the expression of the AKT signal-related protein such as mTOR, pS6 or NF- κ B protein? Certainly in order to show that the signal is regulated, the authors have to show that the related protein is regulated experimentally.

We thank the reviewer for this suggestion. We have performed new experiment in evaluating the protein level of PTEN (new Fig 3g, downregulated in SGS cells), phosphorylated form of mTOR (Fig S4c, upregulated in SGS cells), pS6 (Fig S4b, upregulated in SGS cells; Fig 5f upregulated in mutant organoids), and NF- κ B (Fig S4c, not changed in SGS cells). Moreover, we interfered with AKT over-activation in mutant cells in order to evaluate the phenotype rescue. To do this, we used AKT inhibitor (MK-2206, (Li et al 2017)) which was able to effectively reduce pAKT levels in SGS cells (fig. 3h). We found that the inhibition of this pathway (also pS6 was downregulated at normal levels, Fig. 3i) was enough to normalize the proliferation of mutant NPCs (Fig. 3j). However, we failed to normalize the folding phenotype in organoids possibly due to toxicity of the inhibitors (two different in this case: MK-2206 and GDC-0068) (Fig. for the reviewers 4).

Reviewer #2 (Remarks to the Author):

The manuscript by Banfi et cols presents a study of the molecular mechanisms underlying neurodegeneration in Schinzel-Giedion syndrome (SGS). The authors have generated patient-derived induced pluripotent stem cells (iPSCs) to model neurogenesis in this disorder and performed extensive characterization of the molecular alterations arising during this process. The main conclusions of the study are the following:

- The authors establish a cellular model for SGS using patient-derived iPSCs and isogenic controls, from which neural progenitor cells, cortical neurons and brain organoids are derived.
- The presence of mutations in the SETBP1 degen during neurogenesis leads to an increase in SETBP1 and SET protein in NPCs, inhibiting P53 acetylation and activation.
- NPCs derived from SGS patient iPSCs show accumulation of DNA damage, which is inherited by post-mitotic neurons. This causes increased neuronal death through a PARP1-mediated mechanism.

The study is a detailed analysis of different alterations observed in a neurogenesis model for SGS, providing mechanistic insights into the process of neurodegeneration in this disease (and potentially into options for therapeutic development). It was a pleasure to review this manuscript, as it presents novel and very interesting findings with great relevance for both neurodevelopmental disease and cancer. However, additional data is required to better establish the link between the molecular alterations observed and fully support the molecular mechanisms described in the manuscript.

We thank the reviewer for the kind appreciation of our work. In the new version of the manuscript we added a set of new results in order to better support our conclusions following the reviewer's recommendations:

- a more extensive analysis of the cell cycle alterations adding new flow-cytometry analyses of synchronized cells and an analysis of cell cycle phases in the context of scRNAseq of organoids;
- a thorough investigation of AKT signaling, providing new evaluation of protein levels and using inhibitors to manipulate the signaling;
- a deeper investigation of the cellular phenotype downstream to the P53 upregulation;
- a revised discussion with further elaboration on specific aspects and a generalized revision of the text.

Moreover, we extended the biochemical analyses and the characterization of the brain organoids, including new measurements, scRNAseq and generation of more mature organoids.

Below you can find a point-by-point response to the issues addressed and please note that we highlighted the changes in the manuscript using red sentences.

We hope that the new version of the manuscript has eliminated the weaknesses highlighted by the reviewer

Major comments

1. Figures 2G-H show that SGS NPCs proliferate more than isogenic controls. Additionally, in Figure 2I, the authors show an increase in the proportion of SGS NPCs in S and M/G2 phase and a decrease in the proportion of SGS NPCs in G1 phase compared to isogenic controls. Given that SGS NPCs show: 1) decreased expression for genes involved in mitotic cell cycle arrest and upregulation of genes involved in proliferation (RNA-seq data in Figure 3); and 2) alterations in P53 (Figure 4E), which controls cell cycle arrest, this finding is worth exploring in more detail. For example, the authors could provide additional data to document alterations in progression through cell cycle checkpoints such showing whether SGS NPCs exit G1 phase of the cell cycle faster than isogenic controls (particularly in the presence of unresolved DNA damage).

We thank the reviewer for this suggestion. In the new version of the MS we added a new experiment to analyze G1/S exit where cells were synchronized by double thymidine block and analyzed by flow cytometry at different time points after thymidine release to follow distribution along the cell cycle (G. Chen and X. Deng, 2018). We showed that mutant cells were indeed faster to exit from G1 and to reach the following phases (new Fig. S2h). We also performed scRNAseq in 3D organoids and, interestingly, the distribution of the cell cycle phases suggested high proliferation in accordance with data collected in 2D cellular model (Fig.5i).

Regarding the P53 role on cell cycle alterations, we showed that increasing levels of P53 after the Nutlin treatments were able to restore the normal proliferation of NSCs (see point 4 below).

2. The authors show evidence for dysregulation in the expression of genes involved in the cell cycle (Figure 3), but it is not clear whether this is caused directly by SETBP1 (as a consequence of SETBP1's role in activating gene expression, as described in Piazza et al. 2018), as a consequence of P53 alterations due to inhibition of acetylation or to further downstream effects. Could the authors provide further detail on this?

We fully agree with this reviewer's comment. In fact, it is difficult dissect direct vs. indirect role of SETBP1 in this system. Following the reviewer's suggestion, we performed additional transcriptional analysis on selected cell cycle related genes (*CCND1*, *CCND3*, *CCNG1*, and *MYC*) in NPCs with or without the Nutlin-3a mediated inhibition of P53 degradation (Fig. S6e). This experiment enabled us to show that at least some genes (*CCND1/3*) were at least partially rescued in mutant cells. This left open the question regarding direct/indirect contribution of SETBP1 accumulation for controlling gene transcription at least for the tested genes. Future work will further clarify this aspect.

3. PTEN is shown to be downregulated in Figure S3E and suggested to be connected to an increase in pAKT and hyperactivation of this pathway. The alteration in the AKT/PI3K pathway is significant, as an increase in AKT/PI3K signaling could lead or contribute to inhibition of cell death and increased cell proliferation (as shown in Figure 2G and 2H). As this pathway is involved in various cellular processes in neurogenesis and has been implicated in other neurodevelopmental disorders, it would be valuable to establish this link with further data (e.g. showing a decrease in PTEN at the protein level or an increase in PIP3 levels which would support a link between both observations, examining the effect of AKT signaling on cell proliferation). Given the role of AKT/PI3K pathway in neuronal migration, this finding is also interesting in light of previous results which have shown defects in neuronal migration in mouse models with SETBP1 mutation (Piazza et cols. 2017). Could the authors please comment on this?

We have performed additional experiment in evaluating the protein levels of PTEN (new Fig 3g, downregulated in SGS cells), phosphorylated form of mTOR (Fig S4c, upregulated in SGS cells), phosphor-S6 (Fig S4b, upregulated in SGS cells; Fig 5e upregulated in mutant organoids), and NF-kB (Fig S4c, not changed in SGS cells).

Regarding AKT activation and its possible roles in neuronal migration we have added a new paragraph in the discussion, as suggested.

4. Previous research has shown that SET protein binds P53, hampering its acetylation and activation. SGS NPCs have increased SET protein levels, a decrease in P53 acetylation (Figure S4D) and dampened P53 acetylation in response to DNA damage induced by doxorubicin (Figure 4E). From the data presented, it is not clear whether the disruption to P53 acetylation observed in SGS NPCs causes the other molecular and cellular alterations observed. To examine this, the authors could test whether increasing levels of P53 protein in SGS NPCs restores levels of acetylated P53 protein and consequently other alterations observed in these cells (e.g. cell cycle alterations, correcting the dysregulation of some genes, increasing apoptosis, decreasing PARP-1 activation, etc).

We thank the reviewer to raise this important point. In a new set of experiments, we have analyzed the phenotype of SGS NPCs upon the increase of P53 levels. First, we showed that the ratio of acP53/P53 after Nutlin-3a-mediated total P53 increase, didn't reach normal levels. This was expected since the molecular machinery that cause this impairment (SET accumulation) was still in place. However, we obtained a modest but appreciable increase of acP53 level (Fig. 4f). We suspect that the results were not crystal clear probably inconsequence of the clash of two opposite clues: more net amount of P53 but less P53 induction due to less DNA damage and proliferative activity. However, we were able to demonstrate a decrease of DNA damage (Fig. 4g and S6a), PAR levels (Fig. S11d) and the normalization of the proliferation rate (as number of mitosis as well as growth curve) of the mutant NPCs (Fig. S6b, c). Cleaved-Caspase3 levels were unchanged meaning that apoptosis was not increasing after P53 upregulation, at least in this setting (Fig. S6b, d).

5. The authors generate brain organoids from the SGS IPCs (Figure 5), which reproduce molecular alterations observed in the 2D cell cultures. Furthermore, the brain organoids present an increase in folds comparable to that observed in organoids derived from cells with PTEN loss-of-function. In the paper referenced by the authors (Li et al 2017), the PTEN mutant organoids are treated with inhibitors of AKT activation to determine whether AKT activation is responsible for the expansion and folding phenotypes. Based on these results, it would be interesting to understand the role of AKT

activation in the abnormalities observed in the SGS brain organoids by treating these organoids with inhibitors as described in the above-mentioned paper.

We thank the reviewer for this suggestion. To start with, we interfered with AKT over-activation in mutant cells in order to evaluate the possible rescue of the phenotype in 2D cultures. For this, we used the AKT inhibitor (MK-2206, (Li et al 2017)) which was able to effectively reduce pAKT levels in SGS cells (fig. 3h). We found that inhibiting this pathway (also pS6 was downregulated at normal levels, Fig. 3i) was enough to normalize the proliferation of mutant NPCs (Fig. 3j). However, we failed to normalize the folding phenotype in organoids possibly due to toxicity of the inhibitors (Fig. for the reviewers 4). In fact, when we used MK-2206 as well another AKT inhibitor (GDC-0068) at the concentrations described in Li et al. (2017) during SGS organoid generation we failed to get any organoid alive after the treatments. Thus, we tried to scale down the concentrations of the chemicals to avoid toxicity, but in these conditions the aberrant foldings in SGS organoids was not changed (Fig. for the reviewers 4). We then assume that only treatments for specific time windows will be required to be tested, but this would require an in-depth analysis which is out of the scope of the present work and therefore we decided to not include these data.

6. The authors show that inhibition of PARP-1 reduces DNA damage in SGS NPCs (Figure S7A), which suggests that PARP-1 activation may be contributing to DNA damage potentially as a result of NAD⁺ depletion. This is supported by data in Figure 7D showing decreased NAD⁺/NADH in SGS neurons. Can the authors comment on whether PARP-1 hyperactivation and NAD⁺ depletion could contribute to the severe epilepsy observed in patients with SGS? Furthermore, could the authors comment on the expected effect of PARP-1 inhibition in post-mitotic neurons (i.e. after accumulation of DNA damage during neurogenesis) as a potential therapeutic approach?

We are thankful to the reviewer for this suggestion. We further elaborated on the possible relationship between PARP-1 activity/NAD depletion and epilepsy and its therapeutic potential in the revised discussion, as follow:

“SGS patients suffer of intractable epilepsy that highly contributes to their poor quality of life. The origin of the seizures is currently unknown. One possibility is that the neurodegeneration occurring throughout the cerebral cortex of the patients may lead to unbalanced network activity driving the insurgence of the attacks. Intense PARP-1 activation, through NAD depletion and reduction of SIRT1 deacetylase activity, was indicated as the main cause of neuronal death in an in vitro model of acute acquired epilepsy (Wang et al., Brain research, 2013). Importantly, mutations in the hydrolases ARH3 and TARG, the enzymes involved in removing ADP-ribose modifications, also lead to neurodegenerative diseases presenting epileptic manifestations (Sharifi et al., 2013; Danhauser et al., 2018; Ghosh et al., 2018). Thus, if excessive PARP-1 activity, PAR levels, NAD⁺ depletion and/or parthanatos are responsible for neurological defects seen in SGS patients, either currently available catalytic PARP-1 inhibitors or NAD⁺ repletion, may represent valuable therapeutic options (Wang et al., NeuroReport, 2007; Liu et al., Sci Rep, 2017). However, further studies are necessary to clarify whether such treatments will be effective in post-mitotic neurons that already inherited damage accumulation from the mitotic precursor stages.”

7. It is difficult to follow the relationship between the molecular and cellular abnormalities described by the authors in this manuscript. It would be beneficial to include a drawing or graphical representation to summarize the molecular alterations observed and how they relate to each other.

Also in this case, we would like to acknowledge the reviewer for the suggestion. An illustration summarizing our data is now provided in the Fig S12b.

Minor comments:

1. Please revise the manuscript for language and errors in grammar. Some sentences convey seemingly incorrect information (e.g. p10, line 298: “Of note, the usage of either the broad-spectrum caspase inhibitor carbobenzoxy-valyl-alanyl-aspartyl-[O-methyl]-fluoromethylketone (Z-VAD) or the necroptosis inhibitor Nec-1s had appreciable effects (Figure 7C)” whereas Figure 7C suggests otherwise). Some sentences are difficult to understand (e.g. p18, line 524: “(C) Histograms showing the number of GO categories relative to the indicated different cell types retrieved in genes upregulated in controls (dark green) and mutant NPCs (light green).”). Additionally, there are minor errors (e.g. p2, line 29: “... represent informative cases for shed light on the causative mechanisms.”) and typos (e.g. p29, line 903: “...analyzed by immunofluoresce”).

We thank the reviewer for these observations. We have now corrected these sentences and made a general revision of the text throughout the manuscript.

2. In the manuscript abstract (p2, line 30), it would be preferable to state that SGS is caused by SETBP1 mutations (which then lead to the accumulation of SETBP1 protein), especially since tissue-specific SETBP1 mutations are mentioned further on in the abstract.

We agree and have changed the abstract accordingly.

3. In addition to establishing iPSCs from patients with SGS, the authors derive isogenic control cells by mutating the SGS allele into a wild-type allele. According to Figure 1B and S1A, this was done by targeting the locus with CRISPR and inducing homology directed repair although no information is provided in the Methods on the homologous repair template. Can the authors provide more information on the homologous repair template used (e.g. ssODN, additional plasmid, endogenous copy of the wild-type allele...)? Additionally, the methods state that double antibiotic selection was applied until complete death of non-transfected cells. The assumption is that the antibiotic selection was short to select for transfected cells (rather than for cells with genomic integration of the plasmids), but the Methods would benefit from clarification by specifying the duration of the antibiotic selection.

We apologize for this lack of information. We used homologous repair template (120 bp ssODN) and double antibiotic selection (48 hours). We inserted this information in the methods.

4. Figure 2C shows that NPCs derived from SGS iPSCs have significantly higher SETBP1 protein levels than NPCs from their isogenic controls. However, the graph shows a relatively large bar for the SEM in the quantification of the SETBP1 protein by Western blot visible across the WT, D868N and I871T samples. It would be interesting to understand whether this observation is due to technical/ experimental error (e.g. limitation in the precision of protein quantification with Western blot) or if there are significant differences in the levels of SETBP1 protein in the same cell line across different experiments (e.g. SGS D868N NPCs in experiment #1 versus experiment #2). In relation to this, a question that comes to mind is whether differences in the levels of SETBP1 protein between samples of the same cell line may reflect differences in the level of differentiation of the cells (between iPSCs and NPCs) or differences in the proportion of cells in a specific phase of the cell cycle.

Could the authors comment on this?

This is indeed an interesting observation. Despite we did not perform specific assay to demonstrate that in strong way, we believe that the differences may reflect technical imperfection in the quantification or normal biological fluctuation more than a bias in the iPSC-> NPC differentiation. Indeed, we always perform quality check of our differentiation protocol using immunofluorescence of both iPSC markers (e.g. NANOG, OCT4) and NPC markers (e.g. SOX2, PAX6, FOXG1). However, we cannot exclude that minimal differences in the differentiation state may contribute to the variability in SETB1 accumulation that we reported.

5. In Figure 2I and 3G, results are shown for D868N #1 and #2 which likely represent technical replicates, but this is not stated in the legend or text.

#1 and #2 mean two different clones of the same line. We have stated this in the new version of the manuscript.

6. Figure S1A is too small to read the text. Please make larger or increase the font size. Additionally, Figure S1C shows data for isogenic wild-type cells with mutation I871N. Please correct or clarify.

Thank you for pointing out these problems. We have fixed both issues.

7. Figure S3A shows over- and under-expressed genes in D868N and I871T cells in a Venn diagram. This type of diagram is not optimal to display the data, as there is no overlap expected between over-expressed and under-expressed genes in the same experimental sample.

Although we understand the reviewer's point of view, nonetheless we believe that this illustration is helpful to efficiently show the overall situation without external imposition. Thus, the overall intra-samples (not present) and even the low number of genes that are moving in different directions in one mutant vs the other, depict how the transcriptome of the two lines is. For this reason, we would prefer to maintain this type of representation of the data.

Reviewer #3 (Remarks to the Author):

In this manuscript, Banfi and colleagues study the role of 2 mutant variants of SETBP1, naturally causing Schnitzel-Giedin Syndrom in humans. This is a rare neurodevelopmental disorder that leads to premature neurodegeneration at the early stages. The authors promise in the abstract that studying this specific disease will shed new light on the onset of neurodegenerative diseases, which is not the case. Nevertheless, they characterize the NPCs and neurons derived from these patients, and convincingly show an interesting mechanism underlying this pathology.

I found this manuscript well-written and quite easy to follow. The experiments are justified even though sometimes the logical links between steps are weak.

We are thankful to the reviewer for the positive evaluation of our work. In this revision we added a substantial set of new data in order strengthen our conclusion following the reviewer's recommendations, as in particular:

- adding more methodological details on our experiments (e.g. replicates and numbers);
- performing a new thorough characterization of brain organoids, including new measurements, scRNA-seq and generation of more mature organoids;
- investigating the phenotypes of DNA damage and cell death in neuronal subtypes and astrocytes;

Moreover, we added a revision of the biochemical data, a more precise investigation of the AKT signaling (providing new evaluation of protein levels and using selective inhibitors to manipulate the signaling) and a thorough investigation of the cellular phenotype downstream to P53 upregulation.

Below it is detailed a point-by-point response to the issues addressed and please note that we highlighted the changes in the manuscript using red sentences.

We hope that the new version of the manuscript has solved the weaknesses highlighted by the reviewer.

Specific Comments:

The authors are quite vague on replicates and numbers, they should be more precise about the number of coverslips, technical and biological (batches) replicates, the number of organoids, ventricles, cells analyzed for each experiment since the variability of these systems is very high.

We apologize for this lack of information. In this revision, we added or better specify these methodological data.

I find it quite weird that the genes upregulated in mutant NPCs are associated with different lineages. Did they reproduce the data with different differentiation batches?

Can the authors comment a bit more clearly to which type of cells the mutant NPCs are similar to?

The two different isogenic pairs represent different batches as well. As a matter of fact that this feature is penetrant in both lines, it is already a strong indication. However, we performed validation experiments on the transcriptional levels of genes associated with different lineages (e.g. *THY*, *IL1R1*, *SPP1*, etc) using independent batches of both isogenic lines,

largely confirming previous RNA-seq data (Fig. S4b). Regarding the comment on “which type of cells are similar to”, our computational analysis indicates that they can be still recognized as NPCs, but with the expression of ectopic gene programs that are silenced in the wild type counterparts (this will be the focus of other work of the lab, out of the scope of this work). We revised the discussion on this argument as follow:

“Intriguingly, mutant NPCs up-regulated genes related to cell lineages with very different developmental origin, for instance genes related to the inflammatory response (IL1B, CSF1, ADORA1, IL1R, SPP1) and angiogenesis (THY1, RAMP1, ADM2) among others (Fig. 3c, Supplementary Fig. 3b and Table S2). However, an in silico analysis using t-distributed stochastic neighbour embedding (t-SNE) algorithm indicated that mutant and wild type NPCs correlated with each other more closely and strongly, in respect to other iPSC-derived cell types (Fig. 3d). Thus, these results confirmed that, despite showing a dysregulated molecular signature, mutant NPCs still retained strong neural progenitor identity.”

The organoid data are very vague, how did the authors quantify the “more elaborated” mutant organoids? What does this mean? How did they quantify the folds?

How was the “increase of neuroepithelial like structure” quantified? The number of ventricles? Per organoid? The authors should show the entire organoid with some tissue-clarity method to show this change.

We apologize with the reviewer for the lack of this information in the characterization that remained indeed more qualitative than quantitative. In the new version of the MS we carried out multiple measurements to assess unbiased parameters of control and mutant organoids (see also points below). We measured the circularity (as function of the perimeter and area of the organoid slice = $4\pi(\text{area}/\text{perimeter}^2)$) that confirmed how mutant organoids are farther from circle compared to control counterpart (Fig. 5d). Each organoid was serially sampled evaluating on slice (20 μm) every 140 μm . We believe that our conclusions on the aberrant morphology of the mutant organoids based are now better corroborated with this spatial analysis. In addition, we have quantified the relative number of KI67+ cells on the total DAPI+ nuclei within the rosettes to better circumstantiate our initial qualitative observations regarding the changes in the cell proliferation index (Fig 5e).

How was the neuroepithelial layer expansion assessed? Ventricular length? Thickness? Number of DAPI+ cells at the ventricle?

To quantify this aspect, we performed an in-depth evaluation of the rosettes inside the organoids. We first quantified their number showing that were less in SGS organoids respect to controls (Fig. S7c). However, the mutant rosettes were significantly larger than the control ones, as indicated by the measurements of the luminal (ventricular) length (Fig. S7c). Moreover, we also quantified the thickness of the neuroepithelium showing that on average, the SGS organoids have thick rosettes, in particular due to progenitor area (DCX-) while the differentiated territory appeared thinner (at least in a column perspective) than the control ones (Fig S7d).

Why cortical spheroids did not fold? Are other (non-dorsal) cells necessary for the formation of the folds?

This is an interesting question which, however, cannot be satisfactorily replayed on the basis of the current literature. The suggestion of the reviewer is well taken, but it would require a substantial body of work which remains out of the scope of this manuscript

Where did they show that hyperproliferation occurs in spheroids?

To answer to this question, we now included a proper quantification of KI67+ cells within the rosettes of both control and SGS spheroids (4 weeks in vitro) (Fig. S8c)

The authors then wanted to assess the mutant function in neurons. Why did they artificially express Ngn2? This is not the default protocol to generate neurons, why did they want to speed up the generation of neurons? Can the authors reproduce these data by only changing the medium without forcing the expression of Ngn2 to generate different types of neurons?

We indeed agree with this comment of the reviewer. We initially selected the NGN2 overexpression system since it enabled us to obtain a neuronal culture highly enriched of mature glutamatergic neurons. However, we reproduced similar defects in neuronal shape, pH2AX foci and neurodegeneration in neuronal cultures obtained with a small molecule-based differentiation protocol. However, since the data are perfectly overlapping with the our original protocol, we decided to not include these data in the present MS. We show now the data to this reviewing process with the Fig. for the reviewers 5. However, we have used this differentiation to obtain (some) GABAergic neurons to evaluate in this population the presence of DNA damage and cell death (see below).

Can the authors reproduce the neuronal phenotype in the more physiological 3D organoids?

We presented the WB analysis for pH2AX levels in the original version of the manuscript. In the revision, we added immunofluorescence results on organoids at 1 and 2 months of age showing the increase in DNA damage in both progenitors and post-mitotic neurons (Fig. 5k and S7i). We also performed the TUNEL staining. Since this assay recognizes massive DNA double strand breaks, is presumably a good indicator of DNA damage in addition to highlight dying cells (Fig. S7J).

The authors have invested already in generating cerebral organoids, can they assess the neuronal phenotype in cerebral organoids as well by looking at later stages?

Following this reviewer's suggestion, we added new data gathered older (2 months) organoids in which we were able to show: the maintenance of foldings (qualitative observation - Fig S7H), DNA damage (Fig S7I), and TUNEL assay (Fig S7J).

The function of this protein seems to be quite cell-specific, could the author investigate different neuronal excitatory subtypes, interneurons, glial cells to connect the phenotype to something more concrete with patients?

We thank the reviewer for this suggestion. In the new version of the MS we included analysis of TBR1+ glutamatergic neurons (Fig., S9g), GABAergic neurons (Fig., S9h), and astrocytes (Fig., S10). We described the degeneration as occurring in mutant TBR1+ and GABAergic neurons while astrocytes, despite displayed DNA damage (pH2AX+) accumulation, did not display any sign of cell death. These observations suggest that the degeneration in SGS seems to be restricted to the neuronal populations.

The authors provide evidence that this is the mechanism underlying this neurodegenerative disease with the idea that this could shed new light on others. what is their hypothesis now?

Our work leads to a model in which the neurodegenerative cues originate during embryonic development in progenitor cells and become a cause of death in the descendant post-mitotic neurons. We suggest that at least in some other human genetic conditions featuring early neurodegeneration the dynamics may be conserved. We have elaborated on this aspect in the revised discussion.

I wonder if this hyperproliferation followed by neuronal death is something that can be seen in patients. In light of the massive phenotype regarding proliferation at initial stages, it may be possible to correlate this with an increase in brain size at early stages? Is the correlation with a decrease at later stages?

We share this same curiosity with the reviewer. We indeed actively looked for specific imaging data and human post-mortem tissues (to validate our model of degeneration) without any success. Indeed, the disease is extremely rare and no samples are available worldwide. We are in direct contact with the SGS foundation, a patient organization based in UK, which however does not have available this type of material at the moment. We hope to address better this question in the near future when a mouse model for this disease will be generated and available.

Single-cell RNA-seq in organoids would be the best way to figure out all these details unbiasedly.

We thank the reviewer for this suggestion. We managed to generate scRNA-seq data from our best characterized organoids (Lancaster protocol, 1 month of age). Cells from both genotypes were well intermixed in the uniform manifold approximation and projection (UMAP) suggesting high grade of concordance in their identity (Fig. 5g). We derived clusters that then we organized in categories based on the expression of marker genes (Fig. 5h and Supplementary Fig. 7e, f). On the basis of this annotation, our organoids consisted mainly of progenitor cells (expressing SOX2 and PAX6) as expected after only one month of differentiation (Fig. 5h). Progenitors within SGS organoids eventually express markers of cancer-related proliferative genes (e.g. CRABP2, EGFR, SFRP2) virtually absent or low in the control counterparts (Supplementary Fig. 7g). In particular, SGS organoids were enriched for progenitors with misspecified identity (e.g. cycling progenitors but DCX+ and/or BCL11B+, SLC13A+, NFIB+ etc), immature neurons and depleted of cells from other brain areas (e.g. choroid plexus, diencephalon) compared to control (Fig. 5h and Supplementary Fig. 7f). In accordance with our 2D data (Fig. 2i) the distribution of the cell cycle stages in SGS organoid cells was suggestive of enhanced proliferation, with higher number of cells in S phase and less cells in G1 phase compared to the control condition (Fig. 5i). This accelerated cell cycle dynamics might explain at least in part the excessive folding displayed by the SGS organoids.

Legends Figures for reviewers:

Figures 1-3. Blots used for the quantification in the figures 1c, 1d, 1e, 2c, 2d, 2e, 4c and S11c.

Figure 4. Generation of SGS organoids using AKT inhibitors (MK-2206 or GDC-0068) (a). The usage of the inhibitors at concentration used in Li et. al. CSCs 2016 for normalization of AKT signaling (>75 nM for MK-226 and >750nM for GDC-0068) was toxic for SGS organoids (c) while attempts at low concentration (< 50nM and <500nM) failed to rescue (d) the folding appearance of SGS organoids (b).

Figure 5. A pure small molecules approach (Qi et al., Nat Biotech, 2017) without neurogenin2 overexpression (a), allowed the generation of excitatory neurons with similar yield between control and SGS (b). Both DNA damage (c), neurodegeneration (d) and hypo-maturation (e) were present also using this approach.

Reviewers' Comments:

Reviewer #1:

Remarks to the Author:

Authors addressed most of previous comments. But, I am not convinced with the quantification of the data. Please see below. The phenotypes in NPC, organoids, and NGN2-mediated neurons in SETBP1 mutant seems strong. However, when authors describe the molecular mechanism and quantify the data, some data do not match what is described and quantification does not seem consistent with the given figures.

1. In Figure S4a. It is difficult to see the relative Akt activity is only 2 - 3 fold more when there are no p-Akt bands in the Western blots.
2. In Figure S5d, acTP53 western blot shows at least 5 fold less in D868N. But, the quantification shows less than 1/2.
3. Figure S8b, the size of spheroid seems a lot bigger(at least 2 fold) in D868N, but the quantification seems only 10 - 20% more.

Reviewer #2:

Remarks to the Author:

The authors have done an excellent job in responding to all of the questions raised in the previous round of review. All comments have been thoroughly addressed with an extensive addition of data and I recommend this manuscript for publication.

Three minor comments remain:

- Please revise the manuscript for language and errors in grammar. There are minor errors and typos throughout the text (e.g. legend for Figure 4 on p22 "SETBP1 accumulation causes to P53 hypo-functioning." or p8 line 236 "enhanced levele protected the progenitors from the DNA damage").
- Please revise the manuscript for minor discrepancies in figure numbers (e.g. p12 line 352: "Importantly, the Nutlin-3a treatment that increased P53 levels in NPCs was also able to decrease PAR levels in SGS NPCs (Supplementary Fig. 11c)." should point to Supplementary Fig. 11D; page 7 line 197 "Supplementary Fig. 4a-b" should point to S. Fig. 4a-c)).
- Although Fig.4 for the reviewers (SGS organoids treatment with AKT inhibitors) shows negative results, it would be beneficial to include this figure as a Supplementary figure with the manuscript (e.g. Supplementary Fig. 17).

R. Acuna-Hidalgo

Reviewer #3:

Remarks to the Author:

The revised manuscript by Banfi and colleagues is greatly improved, the authors invested a lot of effort in performing the requested experiments and quantifications and clarified convincingly the points I have raised.

Therefore, I'm fully satisfied with this new version of the manuscript.

Point by Point Response about our MS NCOMMS-20-34839-B by Banfi et al.

Reviewer #1 (Remarks to the Author):

Authors addressed most of previous comments. But, I am not convinced with the quantification of the data. Please see below. The phenotypes in NPC, organoids, and NGN2-mediated neurons in SETBP1 mutant seems strong. However, when authors describe the molecular mechanism and quantify the data, some data do not match what is described and quantification does not seem consistent with the given figures.

We thank the reviewer for the general appreciation of our revised work and for the new suggestions. Here below you can find a point-by-point response to the issues addressed. We hope that the new version of the manuscript has eliminated the weaknesses highlighted by this reviewer.

1. In Figure S4a. It is difficult to see the relative Akt activity is only 2 - 3 fold more when there are no p-Akt bands in the Western blots.

The reviewer is right, the proposed figure is quite misleading with extremely faint bands in pAKT of the control line. We substituted the figure with another WB that better represent the quantification. For clarity and transparency, we also show in the Figure for the Reviewers 6 the blots used in such quantification.

2. In Figure S5d, acTP53 western blot shows at least 5 fold less in D868N. But, the quantification shows less than 1/2. We performed additional experiments that contribute to a new quantification. We now also show in the figure S5d a WB that better represent the quantification of our replicates (around 50% decrease). For clarity and transparency, we also show in the Figure for the Reviewers 6 the blots used in such quantification.

3. Figure S8b, the size of spheroid seems a lot bigger (at least 2 fold) in D868N, but the quantification seems only 10 - 20% more.

We wrongly use an extreme situation, apologize for that. We replaced the figure in the new version of the manuscript with a sample that better represent the data.

Reviewer #2 (Remarks to the Author):

The authors have done an excellent job in responding to all of the questions raised in the previous round of review. All comments have been thoroughly addressed with an extensive addition of data and I recommend this manuscript for publication.

We greatly thank Dr. Acuna-Hidalgo for the general appreciation of our revised work. We apology for the mistakes in the text that we have corrected in the new version of the manuscript.

Regarding the possibility of including the negative data of the AKT inhibitors in organoids: we are cautious in this since we believe that a better titration of the usage of the inhibitors might unmask some effects on proliferation and foldings. Thus, we would opt to not this data because not conclusive on an issue that is out of the scope of our work.

Three minor comments remain:

- Please revise the manuscript for language and errors in grammar. There are minor errors and typos throughout the text (e.g. legend for Figure 4 on p22 "SETBP1 accumulation causes to P53 hypo-functioning." or p8 line 236 "enhanced levele protected the progenitors from the DNA damage").
- Please revise the manuscript for minor discrepancies in figure numbers (e.g. p12 line 352: "Importantly, the Nutlin-3a treatment that increased P53 levels in NPCs was also able to decrease PAR levels in SGS NPCs (Supplementary Fig. 11c)." should point to Supplementary Fig. 11D; page 7 line 197 "Supplementary Fig. 4a-b" should point to S. Fig. 4a-c)).
- Although Fig.4 for the reviewers (SGS organoids treatment with AKT inhibitors) shows negative results, it would be beneficial to include this figure as a Supplementary figure with the manuscript (e.g. Supplementary Fig. 17).

R. Acuna-Hidalgo

Reviewer #3 (Remarks to the Author):

The revised manuscript by Banfi and colleagues is greatly improved, the authors invested a lot of effort in performing the requested experiments and quantifications and clarified convincingly the points I have raised. Therefore, I'm fully satisfied with this new version of the manuscript.

We greatly thank the reviewer for the general appreciation of our revised work.

Reviewers' Comments:

Reviewer #1:

Remarks to the Author:

Authors addressed the comments well and the manuscript is ready to publish.

Point by Point Response about our MS NCOMMS-20-34839-B by Banfi et al.

Reviewer #1 (Remarks to the Author):

Authors addressed the comments well and the manuscript is ready to publish.

We greatly thank the reviewer for the appreciation of our revised work.